# Capturing dynamic phage–pathogen coevolution by clinical surveillance

Yamini Mathur[1,3], Caroline M. Boyd[1,3], Jeannette E. Farnham[1], Md Mamun Monir[2], Mohammad Tarequl Islam[2], Marzia Sultana[2], Tahmeed Ahmed[2], Munirul Alam[2] & Kimberley D. Seed[1✉]

Bacteria harness diverse defence systems that protect against phage predation[1], many of which are encoded on horizontally transmitted mobile genetic elements[2]. In turn, phages evolve counter-defences[3], driving a dynamic arms race that remains underexplored in human disease contexts. For the diarrhoeal pathogen *Vibrio cholerae*, a higher burden of its lytic phage ICP1 in patient stool correlates with reduced disease severity[4]. However, direct molecular evidence of lytic phages driving selection of epidemic *V. cholerae* has not been demonstrated. Here, through clinical surveillance in cholera-endemic Bangladesh, we capture the acquisition of a parasitic anti-phage mobile genetic element, PLE11, that initiated a selective sweep coinciding with the largest cholera outbreak in recent records. PLE11 showed potent anti-phage activity against cocirculating ICP1, explaining its rapid and dominating emergence. We identify PLE11-encoded Rta as the defence responsible and provide evidence that Rta restricts phage tail assembly. Using experimental evolution, we predict phage counteradaptations against PLE11 and document the eventual emergence and selection of clinical ICP1 that achieve a convergent evolutionary outcome. Finally, we discover how PLEs balance their dependence on ICP1 tail proteins for horizontal transmission with the restriction of phage tail assembly by Rta: PLEs construct chimeric tails composed of both mobile genetic element-encoded and phage-encoded proteins to ensure their transmission. Collectively, our findings reveal the molecular basis of the natural selection of a globally important pathogen and its virus in a clinically relevant context.

The infectious diarrhoeal disease cholera, caused by the pathogen *Vibrio cholerae*, remains a major threat to global public health. The Bay of Bengal is considered the source of the continuing seventh cholera pandemic, where seventh pandemic El Tor (7PET) sublineages evolve and spread to vulnerable nations across the globe[5]. Unravelling the factors influencing the evolution and selection of 7PET strains is challenging, as they result from the complex interplay of genetic changes, particularly the flux of new mobile genetic elements (MGEs), and selective advantages that can be difficult to ascertain. Recent metagenomic analyses in cholera-endemic Bangladesh indicate that higher ratios of ICP1, the predominant lytic phage preying on *V. cholerae* in the context of disease[6], correlate with reduced risk of severe disease in patients[4]. This suggests that the acquisition of phage resistance could contribute to outbreak severity and influence the evolution of pandemic lineages. However, direct molecular evidence of this link is lacking, as are mechanistic insights into how the dynamic oscillations of phage susceptibility and resistance unfold in nature.

Among the fluctuating MGEs found in 7PET *V. cholerae* are a family of phage satellites called phage-inducible chromosomal island-like elements (PLEs) that play a crucial role in defending against ICP1 predation[7]. PLEs have evolved intricate defence mechanisms that disrupt the phage life cycle while exploiting phage machinery and structural components to package themselves into modified viral particles, thereby blocking phage transmission[7–10]. PLEs are highly specialized in safeguarding *V. cholerae* populations from predation by ICP1, but phage counteradaptations can neutralize their potent anti-phage activity. Analyses of sparsely collected ICP1 isolates have revealed three anti-PLE mechanisms the phage uses to overcome PLE-mediated hijacking and restore phage propagation. These mechanisms vary among phage isolates[6] and mediate nucleolytic degradation of the PLE genome through distinct mechanisms (Fig. 1a). The phage-encoded origin-directed nuclease (Odn)[11] and attachment-directed inhibitor (Adi)[12] antagonize specific subsets of the ten PLE variants discovered so far[13], targeting sequence variants of the PLE origin of replication and integrase, respectively. A remarkable adaptation in ICP1 is its co-option of CRISPR–Cas, which replaces *odn* in the same genomic locus and provides broad-spectrum counter-defence against all PLEs tested so far[11,14]. Genomic analyses have documented the temporal flux of distinct PLE variants in epidemic *V. cholerae*[13]. However, without knowledge of cocirculating ICP1 genotypes, the molecular factors driving such epidemiological patterns remain unclear. Building on the foundational understanding of PLE–ICP1 conflict, we harnessed high-resolution clinical surveillance in Bangladesh to capture and understand the emergence and selection of epidemic *V. cholerae* and ICP1 in a region critical to the dynamics of pandemic cholera.

[1]Department of Plant and Microbial Biology, University of California, Berkeley, Berkeley, CA, USA. [2]icddr,b, International Centre for Diarrhoeal Disease Research, Bangladesh, Dhaka, Bangladesh. [3]These authors contributed equally: Yamini Mathur, Caroline M. Boyd. ✉e-mail: kseed@berkeley.edu

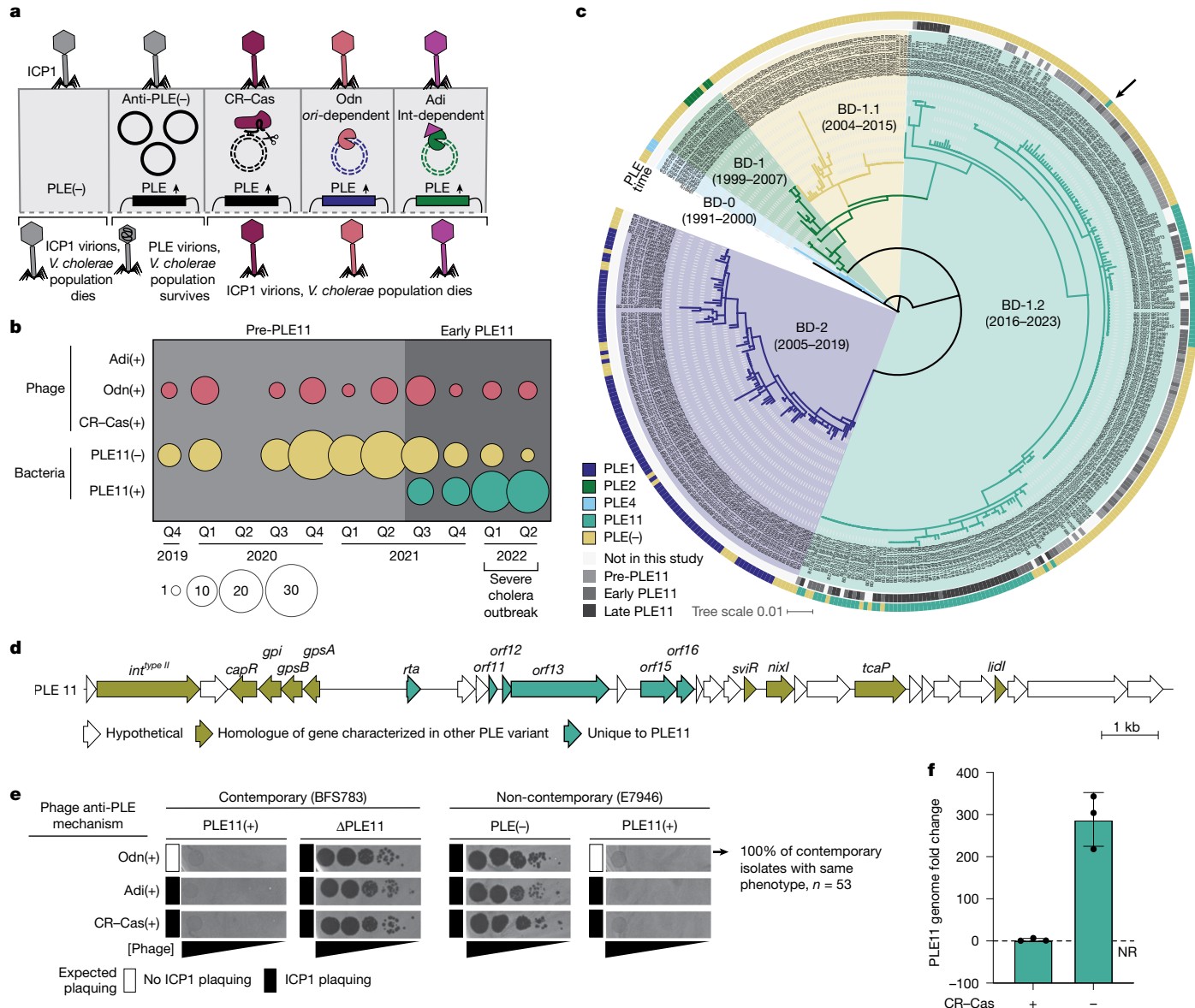

**Fig. 1 | Clinical surveillance reveals the acquisition and selection of PLE11 in *V. cholerae* that restricts ICP1 despite anti-PLE mechanisms. a,** Known infection outcomes of PLE(+) *V. cholerae* with ICP1 isolates encoding anti-PLE mechanisms. Upon infection, PLE excises, replicates and hijacks ICP1 structural proteins for horizontal transmission while abolishing phage production[8–10]. ICP1 counteracts PLE through three mechanisms[11,12,14]: CRISPR–Cas (CR–Cas) cleaves diverse PLE genomes; Odn and Adi antagonize specific PLE variants based on the origin of replication (*ori*) or the presence of a Type II integrase (Int), respectively. **b,** Sampling of 191 *V. cholerae* (*n* = 143 whole-genome sequenced) and 53 ICP1 isolates (*n* = 17 whole-genome sequenced) from stool samples in Bangladesh by genotype and quarter: Q1, January–March; Q2, April–June; Q3, July–September; Q4, October–December. Shading indicates periods before (light grey) and early after (medium grey) PLE11 detection. **c,** Maximum likelihood phylogeny of 431 *V. cholerae* strains from Bangladesh using reference

strain N1961 as an outgroup. Background colours denote lineages, as in ref. 15, with indicated isolation time ranges. Arrow marks the first clinical PLE11(+) isolate (BFS783). Tree scale shows the nucleotide substitutions per site. **d,** Genetic organization of PLE11. Arrows represent genes coloured by functional category. **e,** Plaquing (dark spots) of tenfold serially diluted ICP1 isolates, with the anti-PLE mechanism indicated, on *V. cholerae* lawns (grey background). The expected plaquing phenotypes are indicated. Odn(+) phage on PLE11(+) E7946 is representative of all 53 ICP1 isolates from pre-PLE11 and early-PLE11 periods (Extended Data Fig. 2a). CRISPR–Cas(+) and Adi(+) ICP1 isolates are historically collected isolates. Biological replicates (*n* = 3) are shown in Supplementary Fig. 1c. **f,** PLE11 replication in *V. cholerae* 20 min postinfection with otherwise isogenic CRISPR–Cas(±) ICP1 measured by qPCR. Data are mean ± s.d. (*n* = 3 biological replicates; the dotted line indicates no replication. CR–Cas, CRISPR–Cas; NR, no replication.

## PLE11 fuels *V. cholerae* selective sweep

Between October 2019 and June 2022, we collected and analysed 516 stool samples from patients with suspected cholera from the megacity of Dhaka (*n* = 418) and the small coastal village of Mathbaria (*n* = 98) in Bangladesh. Clinical *V. cholerae* and phage isolates were subjected to phenotypic analysis and genome sequencing to ascertain emerging patterns in their coevolutionary arms race (Supplementary Tables 1 and 2). Notably, this surveillance period coincided with a massive cholera

outbreak in March to April 2022, during which the icddr,b Dhaka hospital treated more than 42,000 patients[15] (Fig. 1b). We sequenced the genomes of 143 *V. cholerae* isolates from unique stool samples during this period as well as 53 from the following 15 months (discussed below) and integrated them with 235 previously published genomes of Bangladeshi *V. cholerae* O1 isolates collected since 1991. Core-genome phylogenetic analysis of the combined genome set (*n* = 431) revealed that strains from this surveillance period belonged to the BD-1.2 sublineage of 7PET wave-3 strains (Fig. 1c). This lineage displaced the previously

locally dominant PLE1(+) BD-2 lineage in 2018 and was initially composed of PLE(−) strains[13,15].

In September 2021, we captured the acquisition of a previously unknown variant of the MGE PLE, designated PLE11, in the BD-1.2 lineage (Fig. 1b,d). Compellingly, PLE11 was acquired in *V. cholerae* amid a backdrop of cocirculating ICP1 isolates invariably encoding Odn as their sole anti-PLE mechanism; further, within 9 months of its initial detection, PLE11 was present in 91% of *V. cholerae* isolates (Fig. 1b). PLE11 does not possess the origin of replication that is sensitive to ICP1's anti-PLE nuclease Odn (Extended Data Fig. 1), indicating this MGE would probably protect *V. cholerae* from cocirculating ICP1 by means of conserved and previously established PLE-encoded defences against ICP1 (Gpi, TcaP, NixI)[8–10] (Fig. 1d and Extended Data Fig. 1). To experimentally evaluate whether PLE11 can protect *V. cholerae* from Odn(+) ICP1 infection, we first used the earliest PLE11(+) clinical isolate detected in our surveillance, BFS783. We constructed a deletion of the PLE in BFS783 and, in parallel, introduced PLE11 into a non-contemporary, phage-sensitive and PLE(−) *V. cholerae* strain, E7946. We then assessed PLE11-mediated inhibition of contemporary ICP1 isolates ($n = 53$), all of which were Odn(+) and recovered only from stool samples with PLE(−) *V. cholerae*. As anticipated, PLE11 was both necessary and sufficient to abolish plaque formation for all cocirculating phage isolates (Fig. 1e and Extended Data Fig. 2a). In comparison, the closely related PLE(−) clinical isolate BFS948 was susceptible to all Odn(+) cocirculating phages (Extended Data Fig. 2b). As such, during the severe cholera outbreak in 2022, no ICP1 isolates were detected that were capable of preying on the dominant PLE11(+) strains that accounted for most of the circulating *V. cholerae*. These data support the model that the MGE variant PLE11 conferred a selective advantage to *V. cholerae* in situ and provide direct evidence that phage resistance contributes to the succession of epidemic strains.

Given the lack of cocirculating phages with an effective means to combat PLE11 during this period, we turned to our historical collection of ICP1 isolates to evaluate predicted infection outcomes based on the known specificity of alternative anti-PLE mechanisms Adi and CRISPR–Cas. We proposed that PLE11 would be susceptible to CRISPR–Cas(+) or Adi(+) phages, given the universal efficacy of CRISPR–Cas against all previously characterized PLE variants[7,13] and its Type II integrase. PLE11(+) *V. cholerae* blocked plaquing even when infected by phages encoding CRISPR–Cas or Adi (Fig. 1e). We omitted the possibility of a PLE11-encoded anti-CRISPR because the PLE11 genome was still degraded upon CRISPR–Cas(+) phage infection (Fig. 1f). This indicated that, unlike all previously characterized PLE variants, PLE11 uses a previously unseen anti-ICP1 defence mechanism that functions despite nucleolytic targeting of the PLE genome. Together, our results show the rapid and dominating emergence of PLE11 that not only protected *V. cholerae* from infection by cocirculating ICP1 but also maintained phage defence in the face of all previously characterized anti-PLE mechanisms.

## Predicting ICP1 counter-defence to PLE11

For ICP1 to sustain its long-term association with epidemic *V. cholerae*, we expected ICP1 to eventually evolve to circumvent defence by PLE11. To predict such an adaptation in nature, we began by investigating the molecular basis of PLE11's unique capacity to restrict ICP1 despite its anti-PLE mechanisms. To this end, we focused on genes unique to PLE11 (Fig. 1d). Deletion of one such gene, *rta*, rendered PLE11(+) *V. cholerae* susceptible to ICP1 infection (Fig. 2a). ICP1 still required an appropriate anti-PLE mechanism to productively infect Δ*rta* PLE11 *V. cholerae*, consistent with PLE11 encoding known defences against ICP1 (refs. 8–10) and PLE11's predicted susceptibility to CRISPR–Cas and Adi (Fig. 2a and Extended Data Fig. 3a). Furthermore, ectopic expression of *rta* alone was sufficient to restrict all 53 contemporary ICP1 isolates to a comparable degree to PLE11 (Extended Data Fig. 2a).

Rta is a small 80 amino acid protein for which we could not identify characterized homologues outside the context of putative satellites

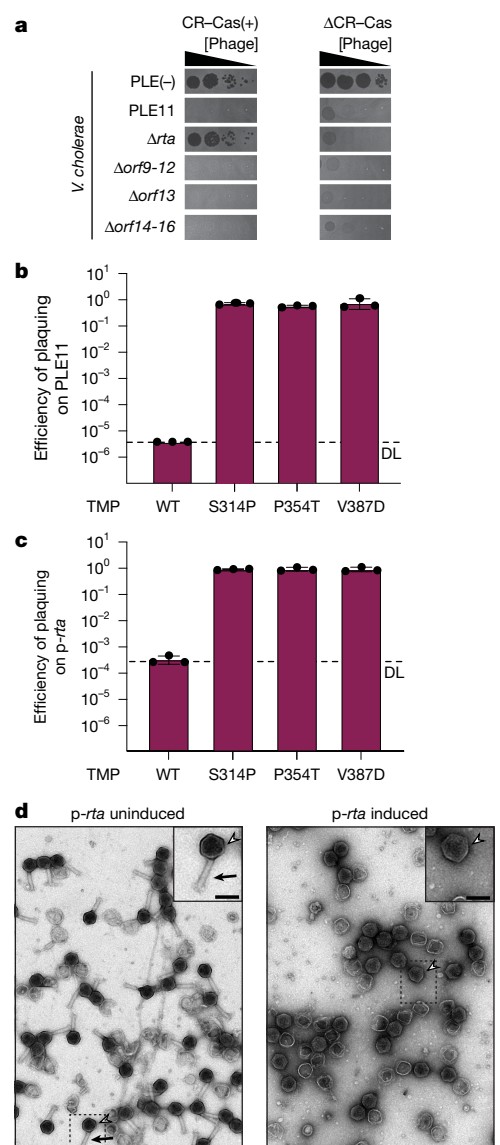

**Fig. 2 | Experimental evolution of ICP1 resistant to PLE11 reveals tail-targeting defence protein, Rta, and predicts counteradaptations needed for ICP1 to persist in nature. a**, Plaquing of tenfold serially diluted ICP1 CRISPR–Cas(±) on lawns of *V. cholerae* strain E7946 and its PLE11(+) and PLE11 mutant derivatives. Biological replicates ($n = 3$) are shown in Supplementary Fig. 2a. **b,c**, Efficiency of plaquing of wild-type CRISPR–Cas(+) ICP1 and experimentally evolved derivatives isolated on PLE11(+) *V. cholerae*, indicated by the relevant substitution in the tape measure protein (TMP), on PLE11(+) *V. cholerae* relative to a permissive PLE(−) strain (**b**) or on Rta-expressing (p-*rta*) *V. cholerae* relative to permissive *V. cholerae* harbouring an empty vector (**c**). Data are mean ± s.d. ($n = 3$ biological replicates). **d**, Representative transmission electron micrographs of particles produced following ICP1 infection of PLE(−) *V. cholerae* with *rta* expressed from a low-copy plasmid (±) inducer ($n = 1$). The arrowheads indicate DNA-filled capsids, and the arrow indicates a virion tail. Further replicates for particles produced following ICP1 infection of *V. cholerae* with empty vector or p-*rta* with inducer are shown in Extended Data Fig. 3e and Supplementary Fig. 2d ($n = 3$ biological replicates). For TEM source data, see Supplementary Fig. 5. DL, detection limit; WT, wild type. Scale bars, 500 nm (zoomed-out views) and 100 nm (insets) (**d**).

(Extended Data Fig. 3b), nor ascertain a potential function using primary sequence or predicted structure. To gain insight into Rta's mechanism of phage restriction in its native genomic context, we evolved CRISPR–Cas(+) ICP1 mutants that escaped PLE11. All escape mutants had single

nonsynonymous mutations in *gp79* or *gp81* (Fig. 2b and Supplementary Table 3), the transcripts of which are spliced to produce ICP1's tape measure protein (TMP)[16]. These phages were also insensitive to ectopic expression of Rta (Fig. 2c), indicating TMP substitutions are sufficient to escape Rta when expressed natively or synthetically. Strengthening these findings, we repeated the experimental evolution approach using a phylogenetically distinct Adi(+) ICP1 isolate and obtained parallel results in which all escape mutants harboured substitutions within the TMP (Supplementary Table 4). By codon swapping *rta* and engineering a splicing-independent *tmp* into ICP1, we further validated that Rta-mediated inhibition is independent of both its nucleic acid sequence and splicing of the TMP transcript (Extended Data Fig. 3c,d).

In contractile phages such as ICP1, the TMP occupies the core of the phage tail and serves as the essential assembly scaffold determining tail length[17]. To substantiate the hypothesis that Rta targets ICP1's TMP, we examined the morphology of ICP1 virions produced from infection of *V. cholerae* ectopically expressing Rta by transmission electron microscopy (TEM). With the induction of Rta, we observed an abundance of genome-filled capsids lacking tails (Fig. 2d and Extended Data Fig. 3e). By contrast, ICP1 derivatives with escape substitutions in TMP produced tailed particles regardless of Rta expression (Extended Data Fig. 3f). Together, these results identify Rta as a new phage defence protein and indicate that Rta disrupts ICP1 infection by targeting its TMP, resulting in defective tailless virions incapable of propagation. Most notably, Rta uniquely enables PLE11 to protect *V. cholerae* populations despite the nucleolytic degradation of the PLE genome mediated by ICP1's anti-PLE mechanisms. Collectively, these data provide a predicted set of adaptations needed for ICP1 to circumvent a recently acquired anti-phage MGE during a natural outbreak of *V. cholerae*.

## Capturing ICP1 counter-defence in nature

Our laboratory evolution studies revealed that ICP1 could evade PLE11-mediated defence through a combination of a nucleolytic anti-PLE mechanism (CRISPR–Cas or Adi) to curb all conserved PLE-encoded defences and a substitution in the TMP to circumvent Rta activity (Fig. 2). Our continued surveillance offered a rare opportunity to evaluate how the genetic basis of adaptation might differ between experimental and clinical phage populations. From July 2022 to September 2023, we collected 189 stool samples to monitor the persistence of PLE11 in *V. cholerae* and track ICP1's evolutionary response to this new mobile phage defence element (Supplementary Tables 1 and 2). PLE11 was detected in 100% (49 out of 49) of BD-1.2 sublineage isolates in 2023, rising from 20% (17 out of 84) in 2021 (Fig. 3a). PLE11(+) strains correspond to two distinct phylogenetic clusters within BD-1.2, suggesting at least two independent acquisitions of this anti-phage MGE (Fig. 1c). The complete replacement of PLE(−) strains with PLE11(+) *V. cholerae* is consistent with the fitness advantage conferred by this mobile phage defence element. By 2023, this genotype shift in *V. cholerae* also coincided with the disappearance of Odn(+) ICP1 phages from stool samples (Fig. 3a). As we observed from the pre-PLE11 and early-PLE11 periods, no Odn(+) isolates from the late-PLE11 period were recovered from patient stool samples in which we isolated PLE(+) *V. cholerae*. These Odn(+) isolates also failed to propagate on PLE11(+) *V. cholerae* and were restricted by Rta (Fig. 3b,c and Extended Data Fig. 6a). Notably, 11 months after PLE11 was initially detected, ICP1 isolates capable of propagating on PLE11(+) *V. cholerae* emerged (Fig. 3a). These phages also showed resistance to Rta-mediated restriction (Fig. 3b and Extended Data Fig. 6a), demonstrating that ICP1 coevolved in nature to circumvent phage defence mediated by the new PLE variant and, specifically, Rta activity.

To identify the genetic basis of these adaptations, we examined the known anti-PLE mechanism(s) in these ICP1 isolates and found they exclusively encoded CRISPR–Cas, with each phage encoding at least one spacer targeting the PLE11 genome (Fig. 3a,c). This genotype shift reflected a replacement of the predominantly clonal Odn(+) ICP1

population with phylogenetically distinct, largely clonal CRISPR–Cas(+) ICP1 isolates (Fig. 3c and Extended Data Figs. 4 and 5). Consistent with observations from experimentally evolved phages, we found that CRISPR–Cas(+) ICP1 isolates cocirculating with PLE11 harboured amino acid substitutions in the TMP (Supplementary Table 2). Mapping the substitutions in evolved phage from clinical specimens revealed they localized within the same 73-amino-acid region of the TMP as identified in our experiments (Fig. 3d). These substitutions, L362P (in all isolates) and N355S (in 79%), were adjacent to residues that conferred escape from PLE11 and ectopically expressed Rta in laboratory conditions, although they were distinct from those arising in experimental conditions (Fig. 3d). To confirm the functional role of naturally evolved TMP substitutions, we engineered L362P or L362P + N355S into the CRISPR–Cas(+) Rta-sensitive historical isolate of ICP1 used in experimental evolution. Both variants were sufficient to evade phage defence mediated by PLE11 through Rta (Fig. 3e,f). Further, reversion of these substitutions in clinical ICP1 demonstrated that L362P was the necessary and sufficient substitution to evade PLE11 and Rta-mediated restriction (Extended Data Fig. 6b,c), demonstrating that ICP1 evolved in nature under selection pressures imposed by a phage defence in *V. cholerae*.

In summary, our surveillance revealed the predicted shift from phages with the ineffective anti-PLE mechanism Odn to phages with CRISPR–Cas that overcomes PLE11's conserved anti-phage defences and substitutions in the TMP to counter PLE11's unique Rta-mediated phage defence. Whereas experimental evolution studies could not select for CRISPR–Cas(+) isolates from an Odn(+) population—due to the lack of pre-existing variation in genetically homogenous stocks—natural populations probably maintain some level of genetic diversity, enabling selection of rare alleles and oscillations of the dominant genotypes. Together, our results document the coevolution of a lytic phage in a clinically relevant context to counter a newly acquired phage defence and demonstrate a convergent evolutionary outcome in nature that mirrors experimental studies.

## PLEs evade Rta by chimeric tail assembly

All well characterized double-stranded DNA phage satellites hijack phage tails for virion assembly[18]. Thus, our data reveal a key paradox: while PLE-encoded Rta interferes with phage tail assembly (Fig. 2), TEM of purified PLE11 virions revealed particles with modified capsids and contractile tails (Fig. 4a). This finding, combined with previous knowledge that the most extensively characterized PLE variant, PLE1, transduces in virions with contractile tails[19] dependent on the ICP1 receptor[7], indicates that PLE11 has a mechanism to assemble tailed virions in the presence of Rta.

As ICP1's TMP is the apparent target of Rta, we proposed that PLE11 may achieve tail formation in the presence of Rta by encoding an alternative, Rta-resistant TMP of its own. Although no other satellites have been reported to encode a TMP, we found that PLE11 encodes a protein with similarity to probable TMPs using HHpred (Fig. 4c). Further, bioinformatic and structural analyses of neighbouring genes revealed that PLE11 also encodes a putative tail assembly chaperone (TAC) and baseplate hub protein (BhuB) (Fig. 4c and Extended Data Fig. 7a). As assembly of the tail tube around the TMP initiates from a multi-component baseplate complex and requires a stabilizing TAC[17], these results indicate that PLE11 encodes a partial tail assembly pathway. Although Rta is unique to PLE11, *tmp* and *tac* genes are conserved across PLE variants, clustering into two sequence groups (exemplified by PLE11 and PLE4, Fig. 4c and Extended Data Fig. 7b). On the basis of the presence of these tail genes, we proposed that PLE11 assembles chimeric tails by hijacking ICP1 components (for example, tail tube, sheath, baseplate components and tail fibres) while substituting the TMP, TAC and BhuB with PLE-encoded proteins (which all have 20%, 20% and 41% identity to ICP1-encoded counterparts, respectively; Supplementary Table 10), thus evading Rta activity.

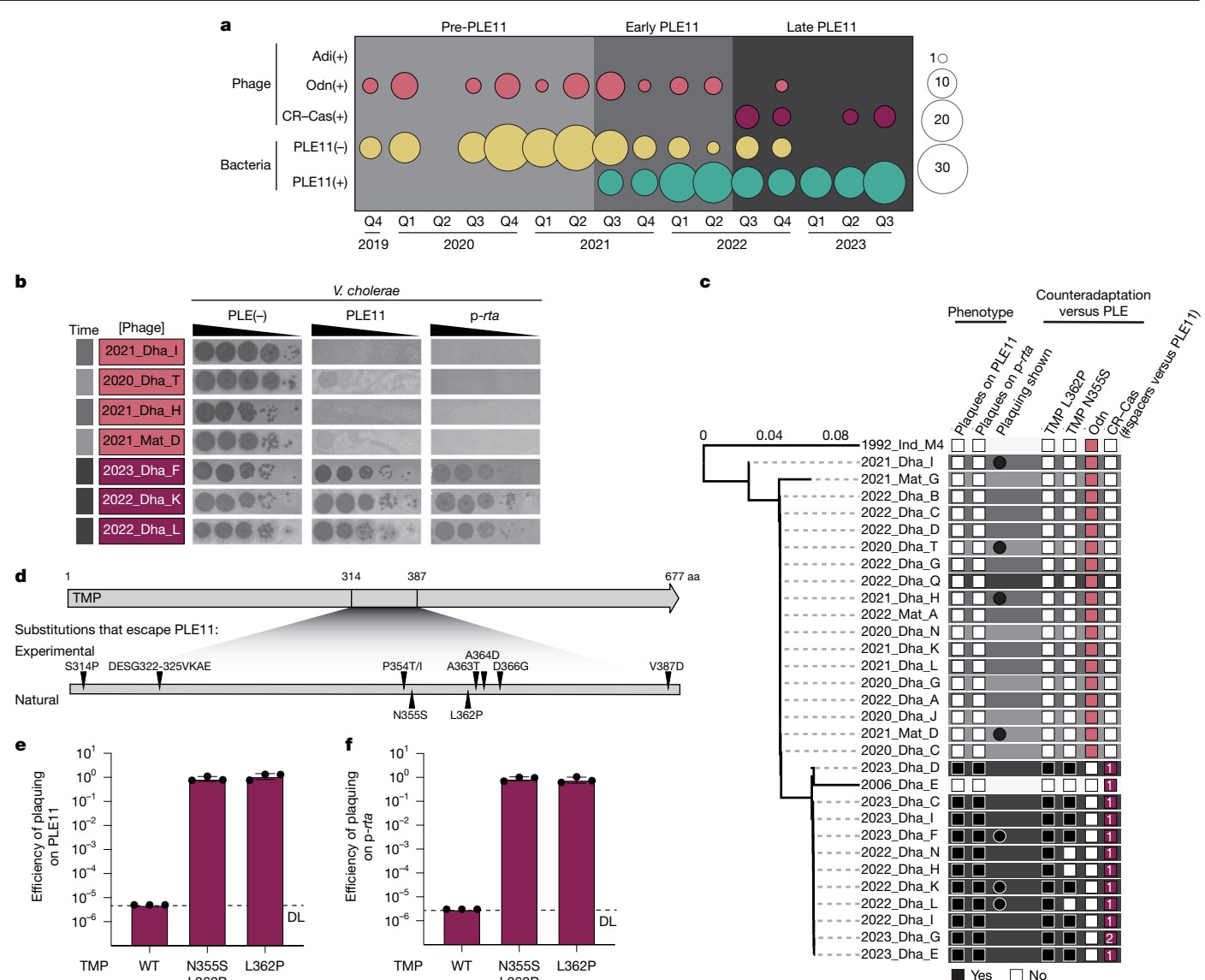

**Fig. 3 | Continued surveillance documents emergence of ICP1 with counteradaptations against PLE11 and Rta. a**, Sampling of 275 *V. cholerae* (*n* = 196 whole-genome sequenced) and 75 ICP1 isolates (*n* = 29 whole-genome sequenced) from stool samples in Bangladesh by genotype. Shading is applicable throughout the figure: pre-PLE11 (light grey), early-PLE11 period (medium grey) and late PLE11 with CRISPR–Cas(+) ICP1 cocirculation (dark grey). **b**, Representative plaquing of tenfold serially diluted ICP1 isolates with the anti-PLE mechanism indicated (Odn, pink; CRISPR–Cas, maroon) on *V. cholerae* strain E7946 and its PLE11(+) and Rta-expressing (p-*rta*) derivatives. For the full panel of phages tested, see Extended Data Fig. 2a (biological replicates (*n* = 3) are shown in Supplementary Fig. 1a) and Extended Data Fig. 6a (biological replicates (*n* = 3) are shown in Supplementary Fig. 3). **c**, Phylogeny of whole-genome sequenced ICP1 phages (*n* = 29 from this study, historical reference

isolates: 1992_Ind_M4 and 2006_Dha_E) through ViPTree. Standardized ICP1 names denote year and isolation location. Empty boxes represent no phenotype or absent genotype, whereas filled boxes show there is a phenotype (yes) or present anti-PLE mechanism (Odn, pink, or CRISPR–Cas, maroon). CRISPR–Cas box numbers: PLE11-targeting spacers (more than 96% identity); 2023_Dha_G carries a new PLE11-specific spacer. **d**, Schematic of ICP1's TMP showing amino acid substitutions. **e,f**, Efficiency of plaquing of CRISPR–Cas(+) wild-type ICP1 and laboratory-engineered derivatives with one or both substitutions seen in the TMP of natural clinical phage isolates cocirculating with PLE11 on PLE11(+) *V. cholerae* relative to permissive PLE(−) (**e**) or Rta-expressing *V. cholerae* relative to empty vector control (**f**). Data are mean ± s.d. (*n* = 3 biological replicates). WT, wild type; DL, detection limit; Dha, Dhaka Bangladesh; Ind, India; Mat, Mathbaria Bangladesh.

To test this hypothesis, we investigated the structural and compositional differences between ICP1 and PLE11 virions. As the TMP determines tail length[17] and the PLE11 TMP and ICP1 TMP differ by 95 amino acids, we expected PLE TMP incorporation to result in roughly 10–15 nm shorter tails[20]. Examination of purified ICP1 and PLE virions by TEM confirmed the expected 10-nm difference between phage and PLE tails (Fig. 4b). Western blot analysis of purified virions confirmed the incorporation of PLE-encoded TMP into PLE11 virions, with exclusion of ICP1's TMP and BhuB (Fig. 4d). To further validate these findings and provide a more thorough evaluation of protein composition, we analysed the virions by

mass spectrometry. For ICP1, we detected all main predicted proteins comprising the capsid and the tail (Fig. 4f and Supplementary Table 5). In PLE11 virions, we detected ICP1-encoded capsid proteins as expected, as well as most of ICP1's tail components, with the notable exception of ICP1's TMP and BhuB. Accordingly, these analyses demonstrated the incorporation of PLE11's TMP and BhuB in PLE11 virions (Fig. 4e and Supplementary Table 5). Two more PLE proteins without predicted functions were also found in the virions (Supplementary Table 5). Collectively, these data demonstrate that PLE11 uses both satellite-derived and phage-derived proteins to assemble functional chimeric tails,

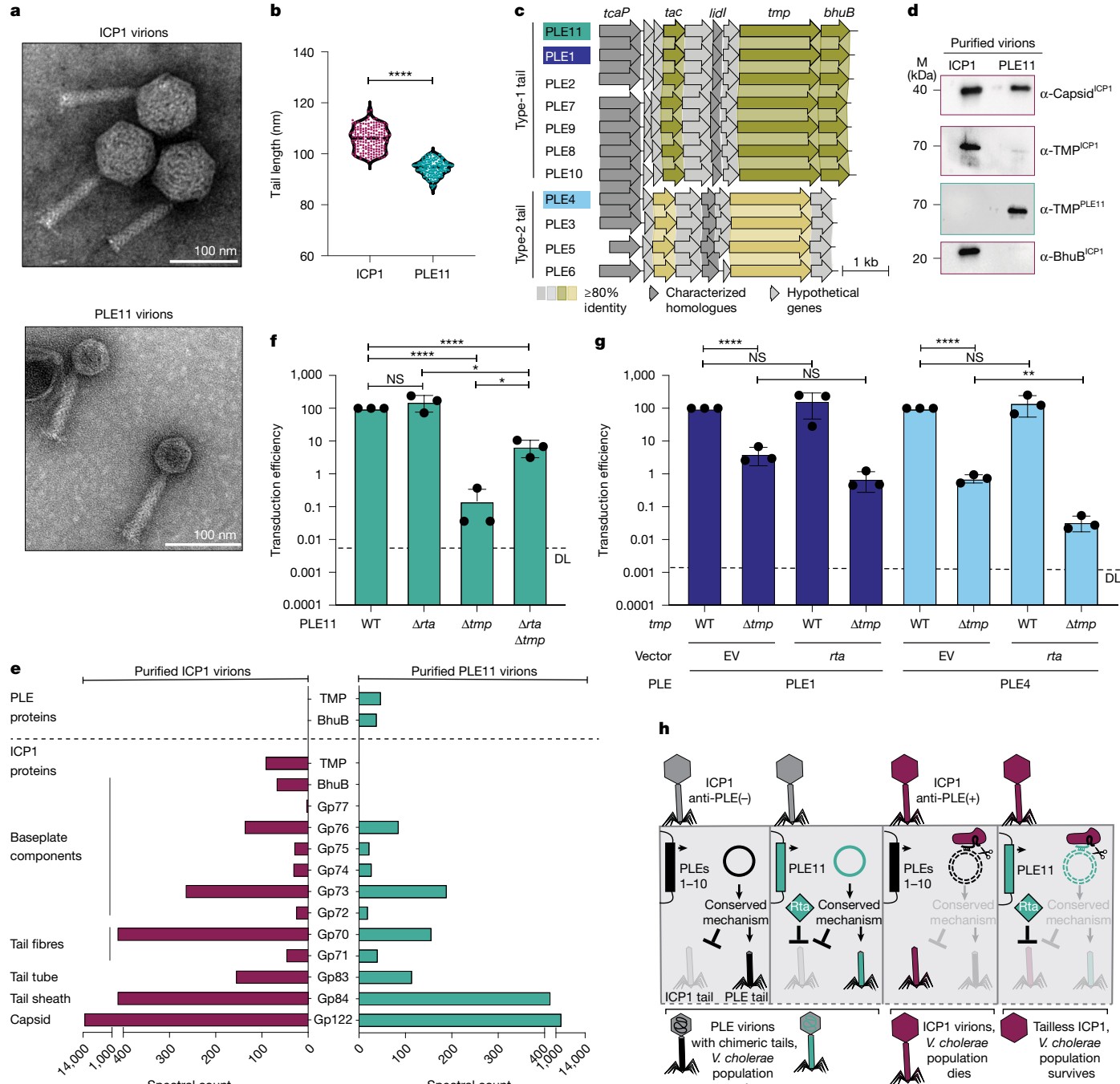

**Fig. 4 | Incorporation of satellite components into chimeric tails by PLEs allows Rta to restrict ICP1's tail assembly. a**, Representative TEMs of ICP1 and PLE11 virions (*n* = 3 biological replicates). For TEM source data, see Supplementary Fig. 5. **b**, ICP1 and PLE11 virion tail lengths from TEM images of purified virions. Dots represent individual particles (*n* = 170 each); horizontal lines indicate means. ****$P$ < 0.0001 (two-sided Student's *t*-test). **c**, Bioinformatic analysis of *V. cholerae* PLE genes (arrows) using tBLASTX. Shading links gene products with more than 80% identity. Previously characterized PLE1 genes are indicated. Homologues of predicted tail proteins (TAC, TMP and BhuB) are grouped by sequence identity (more than 90% within each tail type, less than 25% between tail types; Supplementary Table 10). **d**, Western blot (*n* = 1) of purified ICP1 and PLE11 virions for indicated tail components and ICP1's capsid protein. For gel source data, see Supplementary Fig. 5. **e**, Purified ICP1 and PLE11 virions analysed using mass spectrometry (*n* = 1). The spectral counts (*x* axes) correspond to a subset of proteins (*y* axis) from PLE11 (top) and ICP1

proteomes (bottom) for ICP1 (left) and PLE11 (right) virions. Note that for **d** and **e**, independent preparations of virions were used to corroborate chimeric tail assembly. **f**,**g**, Transduction efficiency of indicated PLE upon infection by ΔCRISPR–Cas ICP1. **f**, PLE11 mutants relative to wild-type control. **g**, PLE1 and PLE4 Δ*tmp* derivatives with an empty vector or Rta-expression plasmid normalized to wild type with the empty vector. Data are mean ± s.d. (*n* = 3 biological replicates). Not significant, $P$ > 0.05; *$P$ < 0.05; **$P$ < 0.01; ****$P$ < 0.0001 (two-sided Student's *t*-test). **h**, Model of PLE-mediated tail manipulation and Rta-mediated anti-phage defence. PLEs encode alternative TMPs that are selectively incorporated into PLE virions through an unknown and probably conserved mechanism. PLE11-encoded Rta, although unnecessary for chimeric tail assembly, protects *V. cholerae* from ICP1 that otherwise circumvents all other known PLE-encoded anti-phage inhibitors (for example, by means of CRISPR–Cas-mediated PLE degradation). EV, empty vector; NS, not significant; *rta*, Rta-expression plasmid; WT, wild type; DL, detection limit.

representing a new mechanism of protein piracy and manipulation among phage satellites that has not been observed previously.

The substitution of PLE11's TMP in the tails of virions provides a plausible mechanism for how PLE11 escapes the inhibition of phage tail assembly imposed by Rta's targeting of the ICP1's TMP. We inferred that PLE11's TMP would thus be essential for its horizontal transduction when Rta was restricting ICP1's TMP. To test this, we compared the transduction efficiency of several PLE11 mutant derivatives with that of the wild-type element. In line with our hypothesis, the deletion of *tmp* in PLE11 caused a notable roughly 99.9% reduction in transduction efficiency, confirming its essential role (Fig. 4f). In a Δ*rta*Δ*tmp* double mutant, transduction was restored significantly, indicating that relief of Rta-mediated restriction of the phage's TMP allows the incorporation of ICP1's TMP into PLE virions. Complementing Δ*rta*Δ*tmp* with ectopic expression of Rta exacerbated the Δ*rta*Δ*tmp* transduction defect (Extended Data Fig. 8), further supporting this model. However, the transduction efficiency of the double Δ*rta*Δ*tmp* mutant remained markedly below wild-type levels (Fig. 4f), suggesting that PLE11 cannot efficiently substitute ICP1's TMP even in the absence of Rta.

The deletion of *rta* alone had no effect on PLE11 transduction (Fig. 4f), indicating Rta's interference with ICP1's tail assembly is not required for incorporating the PLE TMP. Consistent with this, all PLEs encode *tmp* and *tac* genes, suggesting they can produce hybrid tails wherein the PLE-encoded TMP is incorporated. Rta is not part of the PLE tail gene cluster and is uniquely found in PLE11; thus, it is probably not a factor required for the manipulation of tail assembly. In parallel with observations from PLE11, Δ*tmp* derivatives of PLE1 and PLE4, which represent distinct *tmp–tac* gene clusters (Fig. 4c), showed significant transduction defects (96% and 99% reduced efficiency compared with the wild type, respectively) (Fig. 4g), suggesting manipulation of tail assembly to incorporate the PLE TMP is a broad feature of this family of satellites. Ectopic Rta expression had no effect on wild-type PLE1 or PLE4 transduction (Fig. 4g), consistent with these PLEs using their TMP for virion assembly and thus being impervious to Rta's restriction of ICP1's TMP. However, like PLE11, Rta exacerbated the transduction defects of PLE1 and PLE4 Δ*tmp* mutants (Fig. 4g), consistent with the model that PLEs can less efficiently use ICP1's TMP and, when forced to do so, are sensitized to Rta-mediated restriction.

Together, these findings establish that PLEs are the first family of phage satellites known to manipulate viral tail assembly to incorporate satellite-encoded and phage-encoded structural components into chimeric tails. The unique adaptation of Rta encoded by the PLE variant, PLE11, capitalizes on an already existing, but yet unknown, pathway to manipulate phage tail assembly (Fig. 4h). Rta is expressed before ICP1 infection (Extended Data Fig. 9a). Its activity therefore provides robust inhibition of ICP1's tail formation even when the PLE genome is targeted for degradation by ICP1's nucleolytic anti-PLE mechanisms (Fig. 4h and Extended Data Fig. 9b), thus providing a selective advantage to the population when PLE11(+) *V. cholerae* is infected by CRISPR–Cas(+) or Adi(+) phages. These mechanistic insights help to explain the selective sweep of PLE11(+) BD-1.2, sublineage strains during this surveillance period, and the prolonged period for which PLE11(+) *V. cholerae* strains went unchallenged by ICP1.

## Discussion

The interplay between anti-phage defences and the MGEs that encode them has proven far more nuanced than initially envisioned. Phage satellites, for instance, have been shown to shield bacterial populations from phages directly through co-opting a phage's resources that ultimately permit satellite spread or by encoding broad-acting phage defence cargo that is auxiliary to the parasite's horizontal transmission[21]. This former anti-phage activity is typically narrow spectrum, relying on precise molecular interactions between cognate phage and satellite factors. In two ways, our data contribute to an expanded perspective of how satellites defend bacterial populations by manipulating virion assembly and altruistic phage defence. First, it is well documented that phage satellites manipulate capsid assembly and genome packaging[22,23], but PLEs' manipulation of tail assembly to produce chimeric tails reveals a previously unknown aspect of satellite-mediated reprogramming of virion assembly. Although further work is necessary to decipher the benefits of modulating tail assembly, putative satellites found in other *Vibrio* species and unrelated satellites found outside the *Vibrionaceae* also encode predicted *tmp* genes (Extended Data Fig. 10), suggesting that manipulating tail assembly is a broader facet of satellite biology. Second, PLE11's Rta provides the first example of a satellite-encoded defence protein that protects the bacterial population by interfering with the assembly of the parasitized phage independent of satellite transmission. Conversely, other satellites have been shown to encode broad-acting phage defences that spare the phages they parasitize[21], a strategy that presumably benefits both the satellite and the parasitized phage by limiting phage predation of the shared bacterial host. For Rta-mediated defence that restricts phage assembly under circumstances in which the PLE genome is degraded, phage defence is altruistic as it eliminates transmission of both the parasitized phage and the satellite (Fig. 4h). This seemingly selfless defence could be favourable given the clonal populations typical of cholera epidemics, wherein limiting phage predation still ensures vertical transmission of the satellite within the population.

Our results indicate that all PLEs encode Rta-independent mechanism(s) to manipulate phage tail assembly. Although the mechanism(s) remain enigmatic, we speculate that the highly conserved regulatory small RNA, SviR, encoded by all PLEs, contributes to this phenomenon. SviR in the context of PLE1 was shown to bind to transcripts for ICP1's TAC (*gp82*) and BhuB (*gp78*)[24], suggesting it may contribute to decreasing the abundance of these phage factors, favouring chimeric tail assembly and inhibiting phage tail assembly. In support of this prediction, SviR was shown to downregulate levels of ICP1's capsid morphogenesis factor Gp120 during infection of PLE1(+) *V. cholerae*[24] and, notably, Gp120 is absent from PLE11 particles (Supplementary Table 5). Not all PLEs encode an alternative baseplate hub, suggesting that even within this satellite family, PLEs have evolved many strategies to manipulate tail assembly. Satellites' hijacking of phage proteins has provided many examples of distinct mechanisms to disrupt virion assembly of their cognate parasitized phage, and we expect this to continue to expand with the growing appreciation of satellite diversity and abundance[25].

Notably, Rta is unique among PLE-encoded factors that interfere with ICP1's life cycle in its ability to restrict phage assembly even under circumstances in which the PLE genome is degraded. We show that Rta is expressed before ICP1 infection (Extended Data Fig. 9a), enabling its activity to be independent of the fate of the PLE genome. Whether the potent activity of Rta is also due to prolonged protein stability or the specific nature of its targeting of ICP1's TMP remains to be explained, and the lack of recognizable bioinformatic signatures of Rta hinders predictions. Although the selection of numerous escapes localizing to a discrete region of the TMP (Fig. 3d) is suggestive of a direct interaction between ICP1's TMP and Rta, we were unable to detect this interaction using co-immunoprecipitation assays during phage infection. The molecular diversity of phage defence systems is striking, with more than 100 defence systems discovered so far[1]. However, the targeting of tail assembly as a defence strategy has only been discovered very recently[26–28], leaving our understanding of these defence mechanisms in its early stages. With Rta as the fourth example, it will be interesting for future studies to explore the mechanistic comparisons between tail-targeting defences. Furthermore, examining the convergent evolution of diverse satellites to incorporate such inhibitors into their defence arsenal, whereas in some cases developing strategies for manipulating tail assembly, will also be informative.

In Bangladesh, the observed oscillations in the dominance of BD-1 and BD-2 lineages of *V. cholerae*[15] and ICP1 genotypes with unique repertoires of anti-PLE counter-defences[13] are consistent with negative frequency-dependent selection dynamics and point to the maintenance of genetic diversity in under-sampled reservoirs. Such reservoirs may include asymptomatically infected individuals and/or the aquatic environment, although the precise location(s) and degree of diversity within such a reservoir remain unknown. Whereas these oscillations provide a framework for further explanation of the evolution of 7PET *V. cholerae* and its phages, it is challenging to predict widespread dynamics. Although Bangladesh has long been thought to be a part of a wider transmission network for cholera globally[5], surveillance of phages in cholera stool samples is not standard, and the ecological factors[29] influencing the success of a given lineage may vary substantially with each transmission event. Ultimately, our findings underscore the necessity of a holistic approach that combines genomic surveillance with mechanistic insight to understand how emerging lineages successfully spread.

Anti-phage defence systems are seldom studied beyond simplified laboratory conditions, with few mechanisms investigated in their native hosts. This study provides strong evidence that predation by the lytic phage ICP1 can contribute to the selection of 7PET *V. cholerae*. The acquisition of PLE11 into the BD-1.2 sublineage enabled these strains to resist contemporaneous ICP1 phages, probably facilitating their proliferation, and is a likely factor contributing to the unusually devastating cholera outbreak in the spring of 2022 (ref. 15). Notably, PLE11 was also detected in BD-1.2 by another study[30]. This accompanying work suggests that carriage of PLEs may be associated with a trade-off, potentially compromising global outbreak potential. Our findings, when viewed in this context, provide critical mechanistic evidence to support the idea that the intense selective pressure from lytic phage predation may outweigh this potential cost, driving the rapid succession of *V. cholerae* strains in a hyperendemic environment.

Our results provide genotypic and functional evidence to support previous accounts of the assumed role of phage predation in limiting the duration and severity of cholera epidemics[31–33]. In addition, our surveillance highlights the continuing evolution of *V. cholerae* in the Bay of Bengal, emphasizing the importance of sustained genomic and phenotypic monitoring of both phages and bacteria to track the emergence and spread of 7PET *V. cholerae*. Beyond single-nucleotide polymorphism-based phylogenetic analyses of the core genome, our findings underscore the necessity of examining variable accessory gene content to understand the success of emerging lineages and their potential to seed global cholera epidemics. Moving forward, deviations from expected patterns of phage susceptibility could inform public health measures, with phage-resistant linages prioritized as variants of concern. The coevolution of ICP1 to counter bacterial defences highlights the necessity of integrating mechanistic insight from circulating phage genotypes when developing these responses.

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

## Methods

### Statistics and data reproducibility
Where applicable, statistical analyses were conducted using unpaired, two-sided Student's *t*-tests in GraphPad Prism. Relevant statistical results, including *P* values and standard deviations, are reported in the figure legends alongside the data. No statistical methods were used to predetermine sample size, and blinding and randomization were not used.

### Patient stool samples and isolation of bacteria and phage
Cholera patient stool samples were collected and screened for *V. cholerae* and phages as described previously[34]. Stool samples were collected from patients with suspected cholera at the icddr,b Dhaka Hospital and the Government Health Complex in Mathbaria, Pirojpur, under protocol number PR-16083 approved by the icddr,b Ethical Review Committee, with written consent obtained from participants or their guardians. Samples positive for *V. cholerae* serogroup O1 and/or O139 on VC RapidDipStick test (Span Diagnostics) were de-identified and stored at −80 °C with w/v 20% glycerol. For on-site isolation of *V. cholerae*, positive samples were enriched in alkaline peptone water (APW, pH 8.4, Difco) at 37 °C for 6–8 h and then cultured overnight on taurocholate tellurite-gelatin agar (Difco). *V. cholerae* appearing colonies were further confirmed using previously described biochemical and serological methods[35]. Further purification of *V. cholerae* and isolation of phages from stool was performed at the University of California, Berkeley. *V. cholerae* isolates were further purified twice on Luria-Bertani (LB) agar plates and analysed by PCR for PLE and/or whole-genome sequencing (see Supplementary Table 7 for primers). For phage isolation, a panel of *V. cholerae* hosts (including PLE(−) E7946 and PLE11(+) BFS783) was used to probe for phages from stool. Where possible, phages were isolated and purified on the cognate *V. cholerae* strain isolated from the stool sample. Bacterial hosts were grown to the mid-log phase, incubated with a small amount of frozen stool sample collected on a pipette tip (and dilutions thereof) and the mixture was plated in 0.5% LB top agar. Single plaques were picked and purified twice on the same host before being analysed by PCR for *adi*, CRISPR–cas/*odn* or TMP mutations using the primers listed in Supplementary Table 7 and/or whole-genome sequencing.

### Whole-genome sequencing
Genomic DNA from phages and bacteria was purified using Monarch Genomic DNA Purification Kit (New England BioLabs). Phage samples were initially treated with DNase I to remove non-encapsidated DNA. Illumina sequencing (150-base pair by 150-base pair paired end) was performed by the Microbial Genome Sequencing Center or SeqCenter (for all bacteria and most phage), and Nanopore sequencing was performed by the Barker Sequencing Core at the University of California, Berkeley (for a subset of phage isolates). Genomes were assembled using SPAdes[36], and for escape phages selected on PLE11(+) *V. cholerae*, genomes were analysed using BreSeq (v.0.33)[37].

### Bioinformatic analysis
The PLE genomes were aligned on the basis of gene product identity using clinker[38] at a 30% identity cut-off. We performed BLASTn searches against ICP1 genomes using *odn*, *adi* and CRISPR–cas from ICP1_2001_Dha_0, ICP1_2006_Dha_E or ICP1_2011_Dha_A as queries. TMP substitutions were called if the sequence differed from ICP1_2006_Dha_E or ICP1_2011_Dha_A. ICP1 CRISPR spacers were manually annotated between direct repeats in the CRISPR arrays. The phage phylogeny was built by comparing whole-genome sequences of 29 phages isolated from this study and 67 isolates from previous work[6] using tBLASTx analysis from ViPTree[39]. The intergenomic similarities between phages sequenced in this study were determined at VIRIDICweb using BLASTn parameters '-word_size 7 -reward 2 -penalty -3 gapopen 5 -gapextended 2'

(ref. 40). ICP1 and PLE11 gene products identified in proteomics were analysed for functional predictions using HHPred[41] or extracted from previous work (for ICP1)[6].

The phylogeny of bacterial genomes was calculated as described previously[15]. Briefly, fastp v.0.23.2 (ref. 42) was used to evaluate the quality of the raw shotgun paired-end sequences. Genetic variants were identified by mapping the raw reads to the *V. cholerae* N16961 reference genome (National Center for Biotechnology Information (NCBI) accession IDs NC_002505.1 and NC_002506.1) using snippy v.4.6.0 (ref. 43). Phylogenetic analysis was performed using IQ-TREE v.2.2.0 (ref. 44) with 1,000 bootstrap and the best fitted evolutionary model was selected using ModelFinder[45]. Spades v.3.15.4 genome assembler was used to generate contigs. Each of the ten previously known PLEs[13] and PLE11 were used as BLASTn queries against the *V. cholerae* genomes and annotated in the phylogeny. Lists of the single-nucleotide polymorphisms in the core genome and strains used to build the phylogeny are in Supplementary Tables 8 and 9, respectively.

The structural predictions for TAC$^{PLE4}$ and Rta were made using Colab-Fold[46] on COSMIC2 (ref. 47) and GoogleColab Structural alignments TAC$^{HK97}$ (Protein Data Bank (PDB) ID 2OB9), TAC$^{PLE1}$ (5IR0) and predicted TAC$^{PLE4}$ were done on ChimeraX[48] using 'Smith-Watermann ssFraction 0.8008 matrix BLOSUM-45 hgap 10 sgap 10 ogap 4' parameters. The root mean-squared deviation values for aligned pruned amino acid residues are reported. Putative satellite genomes from non-cholera *Vibrio* spp.[34] and cf-PICIs[49] were analysed for the presence of TMPs and integrases using BLASTp and HHPred[41]. Genome visualizations were generated with R, using the gggenes package.

### Bacterial growth and cloning
Bacterial and phage strains are reported in Supplementary Tables 1, 2 and 6. *V. cholerae* cultures were grown in LB medium at 37 °C with aeration or on LB agar plates. When required, antibiotics were used at the following concentrations: 100 μg ml$^{-1}$ spectinomycin, 75 μg ml$^{-1}$ kanamycin, 2.5 μg ml$^{-1}$ chloramphenicol in LB plates and 1.25 μg ml$^{-1}$ chloramphenicol in liquid cultures. *Escherichia coli* cultures were grown in LB medium at 37 °C with aeration or on LB agar plates, and 25 μg ml$^{-1}$ chloramphenicol was when needed. *V. cholerae* strains were made naturally competent through previously reported methods[50] and then were transformed with DNA fragments generated from purified PCR products. To create the laboratory strain of PLE11(+) *V. cholerae*, a *kan$^R$* cassette was inserted downstream of the last PLE11 ORF by means of the natural transformation of clinical strain BFS783. The PLE11:*kan$^R$* was then transduced into *V. cholerae* E7496. Gene deletions in the PLE11 *kan$^R$* E7946 derivative were made by replacing the target gene(s) with a *frt-spec$^R$-frt* cassette by natural transformation, and where indicated, the *spec$^R$* was cured as described previously[10]. To generate the plasmid p-*rta*, the *rta* coding sequence from PLE11 was cloned into the pKL06.2 vector using Gibson assembly and selected in *E. coli* XL-1 Blue. The purified plasmid was electroporated into *E. coli* S17 and conjugated into *V. cholerae* strains. All deletions and plasmid constructs were confirmed by PCR and Sanger sequencing.

### Phage propagation, generation of escapes and phage engineering
For standard plaque assays, phage isolates were propagated on *V. cholerae* to generate confluent lysis plates, collected in STE buffer (100 mM NaCl, 10 mM Tris-Cl pH 8.0, 1 mM EDTA), treated with chloroform and clarified by centrifugation at 5,000*g* for 15 min. The aqueous layer containing phages was stored at 4 °C. Escape phages were generated by picking individual plaques from plaque assays with the parental phage, ICP1_2006_Dha_E (for CRISPR(+)) or ICP1_2011_Dha_A (deleted of CRISPR spacers targeting PLE11 (ACMphi232), for Adi(+)), on PLE11(+) *V. cholerae* (JEF42). The plaques were purified three times, and then genomic DNA was extracted from high-titre stocks and whole-genome sequenced (ICP1_2006_Dha_E), or the *tmp* was analysed by Sanger

sequencing (ICP1_2011_Dha_A derivatives). Phages with engineered mutations were generated by passaging the parental phage on a *V. cholerae* strain expressing an inducible CRISPR–Cas system with a targeting spacer and a repair template encoding the desired mutation as described previously in ref. 51. Individual plaques were purified, and the *tmp* sequence was confirmed by Sanger sequencing.

## Phage assays

Saturated *V. cholerae* cultures (optical density at 600 nm ($OD_{600}$) greater than 2) were diluted to $OD_{600} = 0.05$ and grown to $OD_{600}$ between 0.3 and 0.5 (mid-log) at 37 °C with aeration. Cultures were induced with 1 mM isopropyl-β-D-thiogalactoside (IPTG) and/or 1.5 mM theophylline at $OD_{600} = 0.2$ when necessary. For spot assays, 200 µl or 300 µl of mid-log *V. cholerae* cultures were mixed with 0.5% LB top agar (with required antibiotics and inducers, as needed) and poured on 100-mm or 150-mm plates, respectively. Next, 4 ml or 12 ml of this mix was poured onto 100-mm or 150-mm LB agar plates, respectively. Tenfold serial dilutions of phage were prepared, and 3 µl of each dilution was spotted onto the prepared top agar with bacteria, allowed to dry and incubated at 37 °C for 6 h. To quantify plaquing efficiencies, 10 µl of phage dilutions were mixed with 50 µl of mid-log *V. cholerae* for 7–10 min at room temperature. The phage-bacteria mix was added to 0.5% LB top agar (4 ml) in 60-mm plates. Inducers and antibiotics were added when required. The plates were incubated at 37 °C for 6–7 h and then at room temperature overnight. The efficiency of plaquing was calculated from three biological replicates relative to the permissive empty vector control (for p-*rta*) or permissive PLE(−) control (for PLE11(+)). Data presented as images represent one of the three biological replicates, and extra replicates are reported in Supplementary Information.

## qPCR

PLE11 replication PCRs were performed as described in ref. 12. Briefly, saturated *V. cholerae* cultures were diluted to $OD_{600} = 0.05$ and grown to $OD_{600} = 0.3$ at 37 °C with aeration. Immediately before infection ($t = 0$), 100-µl samples were collected and boiled. The remaining culture was infected with phage (ICP1_2006_Dha_EΔCRISPRΔcas2-3 or wild-type ICP1_2006_Dha_E) at a multiplicity of infection (MOI) of 2.5, incubated at 37 °C with aeration for 20 min ($t = 20$), collected and boiled. PLE11 DNA from boiled samples was measured in technical duplicates by quantitative PCR (qPCR) relative to a standard curve using primers indicated in Supplementary Table 7. The fold change in genome copy was calculated as the amount of PLE11 genome at $t = 20$ relative to the PLE11 genome at $t = 0$. qPCR experiments were performed with three biological replicates.

## ICP1 propagation on *V. cholerae* with p-*rta* or PLE11 derivatives

ICP1 production in the presence of an inducible PLE gene was done as described previously in ref. 10. An overnight culture of the relevant *V. cholerae* strain was back diluted to $OD_{600} = 0.05$ in 50 ml of LB medium and incubated at 37 °C with aeration. For *V. cholerae* strains containing either pEV or p-*rta*, 1.25 µg ml⁻¹ chloramphenicol was supplemented in media and at $OD_{600} = 0.2$, 1 mM IPTG and 1.5 mM theophylline were added for induction, or an equivalent volume of sterile water was added in the uninduced control. All cultures were grown to $OD_{600} = 0.3$. and then infected with ICP1 strains at an MOI of 2.5. On lysis (roughly 90 min), the culture was treated with 0.25 units ml⁻¹ benzonase and 10% chloroform for 5 min with shaking, centrifuged at 5,000$g$ for 15 min at 4 °C. The supernatant was centrifuged at 26,000$g$ for 90 min at 4 °C. The resulting phage pellet was resuspended in phage buffer (50 mM Tris–HCl pH 8.0, 100 mM NaCl, 10 mM MgSO₄, 1 mM CaCl₂) by rocking overnight at 4 °C, treated with chloroform (1:1) and the aqueous layer was collected for further analysis. For TEM analysis of ICP1_2006_Dha_E_ΔCRISPR-Δcas2-3 propagated in the presence of p-*rta*± inducer (Fig. 2d), one biological replicate was performed; three biological replicates of TEM analysis of this phage propagated in the

presence of an induced empty vector or induced p-*rta* are presented in Extended Data Fig. 3e and Supplementary Fig. 2d. For TEM analysis of ICP1 2006_Dha_E derivatives with TMP substitutions, phages were propagated in the presence of p-*rta*± inducer (Extended Data Fig. 3f), one biological replicate was performed for each of the three mutant derivatives. For TEM analysis of particles produced by ICP1 CRISPR–Cas(+) on infection of *V. cholerae* PLE11 or PLE11Δ*rta*, three biological replicates were performed as shown in Extended Data Fig. 9b and Supplementary Fig. 4.

## Purification of ICP1 and PLE11 virions

ICP1 and PLE11 virions were generated by infecting 200 ml of mid-log culture of PLE(−) *V. cholerae* and PLE11(+) *V. cholerae*, respectively, at MOI of 2.5 with ICP1_2006_Dha_EΔCRISPRΔcas2-3. After culture lysis, 0.25 units ml⁻¹ benzonase and 10 ml of micellar mix (4 ml of chloroform, 2 ml of methanol, 25 mM sodium citrate and 10 mM sodium deoxycholate) were added and mixed for 5 min at room temperature. The lysate was centrifuged at 5,000$g$ for 10 min at 4 °C; the aqueous supernatant was centrifuged at 26,000$g$ for 90 min at 4 °C. The supernatant was discarded, and the pellet was recovered in phage buffer by rocking overnight at 4 °C. The resuspended pellet was chloroform treated (1:1), and the aqueous layer containing virions was collected. Next, to separate particles from aggregates and free proteins, the concentrated phage stock was subjected to bench-top gel filtration chromatography as follows: 3 g of BioRad P-10 Gel Fine 45-90 was hydrated in 50 mM Tris-Cl pH 8.0 at room temperature and then stored at 4 °C until the next steps. BioRad Polyprep chromatography columns were packed with roughly 3 ml of the hydrated gel to a 1.5-inch bed height. The gel was equilibrated with 15 ml of phage buffer supplemented with benzonase (0.25 units ml⁻¹) using gentle syringe pressure. The concentrated phage stock (roughly 400 µl) was loaded on the resin under gravity, followed by roughly 3 ml of phage buffer with benzonase. The eluent was collected as 100 µl × 30 fractions, which were screened for virions using spot plate plaque assays for ICP1 and transductions for PLE11. The fractions with high titres were also visually inspected using TEM.

## TEM

To stage the virions for imaging, copper mesh grids (Formvar/Carbon 300, Electron Microscopy Sciences) were loaded with 5 µl of samples for 1 min, washed with 5 µl of sterile ddH₂O for 15 s and stained with 1% uranyl acetate for 30 s. The absorbent paper was used to wick liquids between each step gently. The grids were imaged on a Tecnai-12 electron microscope at 120 kV. Uncropped TEM images can be found in Supplementary Fig. 5.

## Particle measurements

The dimensions of the tails of ICP1 and PLE11 virions were measured using TEM images analysed using Fiji[52]: the pixel distance was set to a scale distance (nm) using 'Set Scale'. Tail sizes were measured in a straight line from neck to base using the 'Analyze' > 'Measure' option. Particle measurements were performed on three biological replicates of particle purifications, for a total of $n = 170$ particles measured for each ICP1 and PLE11.

## Mass spectrometry

Three of the highest concentration fractions from gel filtration were pooled (totalling $4.95 × 10^{12}$ TFU per ml for PLE11 virions and $3.00 × 10^{11}$ PFU per ml for ICP1 virions: ICP1_2006_Dha_EΔCRISPRΔcas2-3), and each pool was denatured in Lamelli buffer. Then 40 µl of each sample was run on Any-Kd Mini-PROTEAN TGX Precast gel (BioRad). The gel was stained with GelCode Blue stain (Thermo Fisher) for visualization, and the lanes with samples were cut into 1-mm² pieces and destained. In-gel digestion and mass spectrometry analysis were conducted at the Vincent J. Coates Proteomics/Mass Spectrometry Laboratory at the University of California, Berkeley, USA. Briefly, gel pieces were washed

twice with 50% acetonitrile (ACN) and 50 mM ammonium bicarbonate for 15 min with shaking and then dehydrated with 100% ACN for 5 min, followed by air drying for 5 min. Pieces were further dried with 10 mM Tris(2-carboxyethyl)phosphine hydrochloride and 40 mM chloroacetamide for 5 min at 70 °C. The gel was rehydrated in 50 mM HEPES pH 8.0 in a minimal volume enough to cover the surface. For digestion, rehydrated pieces were incubated with 1 µg Trypsin (1:50 dilution) for 1 h at room temperature, then supplemented with a minimal volume of 50 mM HEPES pH 8.0 and incubated overnight at 37 °C. Peptides were extracted stepwise in treatment with 25% and 100% ACN. Peptide extracts were concentrated to 30–60 µl and acidified with formic acid.

The digested peptides were analysed by online capillary nano liquid chromatography with tandem mass spectrometry using a 25 cm reversed-phase column fabricated in-house (50 µm inner diameter, packed with ReproSil-Gold C18-1.9 µm resin, Dr. Maisch) equipped with laser-pulled nanoelectron spray emitter tip. Peptides were eluted at 100 nl min$^{-1}$ on a Thermo Fisher Easy-nLC1200 using a 140-min linear gradient of 2–40% buffer B (buffer A, 0.05% formic acid in water; buffer B, 0.05% formic acid in 95% ACN in water). Peptides were ionized using a FLEX ion source and analysed on a Fusion Lumos Tribrid Orbitrap Mass Spectrometer (Thermo Fisher Scientific), and data were acquired in Orbitrap mode with parameters as follows: MS1 resolution of 120,000 at 200 $m/z$ and scan range of 350–1,600 $m/z$. The top 20 most abundant ions were fragmented through collision-induced dissociation with 35% normalized collision energy, activation $q$ of 0.25 and a 2 $m/z$ precursor isolation width. Dynamic exclusion was set to a 30-s repeat duration, 20-s exclusion and a single repeat count. RAW files were analysed with PEAKS (Bioinformatics Solutions Inc.) using semi-specific cleavage at R (Arg) and K (Lys) (up to 4 missed cleavages), a precursor mass tolerance of 15 ppm and fragment mass tolerance of 0.5 Da. Variable modifications included methionine oxidation, and cysteine carbamidomethylation was fixed. Peptide hits were filtered using a 1% FDR, with proteins requiring at least 2 unique peptides and 1% FDR. Label-free quantitation was performed with PEAKS using default parameters, except for selecting the top 2 peptides per protein with a minimum abundance of $10 × 10^4$ and normalization based on total ion chromatogram across technical replicates. Data presented in Fig. 4e are based on one biological replicate (note that the western blot analysis of particles was performed on a separate biological preparation of particles).

### Custom antibodies and western blotting

The antibody against the ICP1's capsid protein ($\alpha$-capsid$^{ICP1}$) was used as described in ref. 19. Polyclonal antibodies against ICP1's Gp78 ($\alpha$-BhuB$^{ICP1}$), ICP1's TMP ($\alpha$-TMP$^{ICP1}$) and PLE11's TMP ($\alpha$-TMP$^{PLE11}$) were generated in rabbits by GenScript. Given the intrinsic toxicity and challenges in expressing TMPs, regions of least relative disorder were predicted using the modelled structure generated by ColabFold[46] and sequence-based predictions from DEPICTER2 (ref. 53). The codon-optimized DNA sequence for M252–A338 from ICP1's TMP and M205–N319 from PLE11's TMP were cloned in pET30a(+), 6×-His-tagged proteins were purified using Ni-affinity chromatography and used as antigens.

To analyse the presence of listed proteins in purified virions, samples were prepared in Lamelli buffer, denatured and loaded onto SDS–PAGE in duplicate. The proteins were transferred to 0.22 polyvinyl difluoride membrane using TransBlot Turbo (BioRad) at 1.5 V for 7 min, and the blot was blocked with 0.5% skim milk in Tris-buffered saline with Tween (TBST) (20 mM Tris-Cl pH 7.6, 150 mM NaCl 0.1% Tween 20). The blot was cut at 50 kDa and 25 kDa. The pieces larger than 50 kDa were probed with $\alpha$-TMP$^{ICP1}$ and $\alpha$-TMP$^{PLE11}$ (diluted 1:1,500), blots from 50 kDa to 25 kDa were probed with $\alpha$-capsid$^{ICP1}$ (diluted 1:1,500) and pieces smaller than 25 kDa were probed with $\alpha$-BhuB$^{ICP1}$ (diluted 1:3,000). The primary antibodies were diluted in TBST with 2% bovine serum albumin. Blots were incubated in primary antibodies overnight at 4 °C, washed three

times in TBS (20 mM Tris-Cl pH 7.6, 150 mM NaCl) and incubated in goat anti-rabbit secondary antibodies in TBST with 2% bovine serum albumin for 45 min, and then washed twice in TBS. Blots were developed in ECL chemiluminescence reagent (BioRad) and imaged on BioRad Chemi-Doc. Data presented in Fig. 4d are based on one biological replicate (note that the mass spectrometry analysis of particles was performed on a separate biological preparation of particles). Uncropped western blot images can be found in Supplementary Fig. 5.

For detection of Rta-3xFLAG, overnight cultures of *V. cholerae* PLE11 or PLE11:*rta*-3xFLAG were diluted to $OD_{600} = 0.05$ in 50 ml of LB medium and grown to $OD_{600} = 0.3$, 25 ml of the cultures were retrieved into equal volumes of chilled methanol. The remaining 25 ml of the cultures were infected with either ICP1_2006_Dha_E CRISPR–Cas(+) or ICP1_2006_Dha_E_$\Delta$CRISPR$\Delta$cas2-3 at an MOI of 2.5. At 16 min postinfection, the cultures were collected into equal volumes of chilled methanol. Cells were gathered at 5,000$g$ for 10 min at 4 °C and washed with 1× PBS. Cells were lysed in buffer containing 50 mM Tris-Cl, 5 mM SDS and 1 mg ml$^{-1}$ lysozyme. Total protein estimation for all samples was done using Pierce BCA protein assay kit. For western blot analysis, 70 µg of total protein was loaded onto SDS–PAGE gel and, after transfer, the blot was cut below 25 kDa, then probed with $\alpha$-FLAG (GenScript, diluted 1:2,000) and further probed with goat anti-rabbit secondary antibodies as described above.

### Transduction assays

Donor strains of *V. cholerae* with PLE marked with a $kan^R$ gene downstream of the last PLE gene were grown to saturation. The donor strains were back diluted to $OD_{600} = 0.05$ in 2 ml of LB medium, grown to $OD_{600} = 0.3$ and infected with ICP1_2006_Dha_E_$\Delta$CRISPR$\Delta$cas2-3 at MOI of 2.5. After 5 min, the unbound phage was washed off, and the infected cells were resuspended in fresh media. After roughly 20 min, the lysates were collected, treated with 20 µl of chloroform and centrifuged at 5,000$g$ for 15 min at 4 °C. As necessary, 1.25 µg ml$^{-1}$ chloramphenicol was added to the media for the growth of donor strains but was excluded from the washing and resuspension steps. Strains harbouring pEV or p-*rta* were induced at $OD_{600} = 0.2$ with 1 mM IPTG and 1.5 mM theophylline. In parallel, the recipient strain, *V. cholerae*($\Delta$*lacZ::spec*), was grown for 6–7 h and supplemented with 10 mM MgSO$_4$ right before adding the lysate for transductions. Next, 20 µl of supernatant was mixed with 180 µl of the recipient. Tenfold dilutions of the mix were prepared in LB medium and incubated at 37 °C with shaking for 20 min. All dilutions were then plated on LB agar plates containing kanamycin and spectinomycin, and the resulting colonies were counted to calculate transducing-forming units per millilitre of the donor lysate. The transduction efficiency was calculated relative to wild-type PLE, wild type with empty vector or empty vector controls as applicable. Transduction efficiency was calculated from three biological replicates.

### Reporting summary

Further information on research design is available in the Nature Portfolio Reporting Summary linked to this article.

## Data availability

Sequence data for *V. cholerae* and ICP1 isolates from clinical samples and for ICP1 escape phages have been deposited in the NCBI Sequence Reads Archive under BioProject PRJNA1195958. The PLE11 sequence has been deposited to GenBank (accession no. PQ783903). The mass spectrometry proteomics data have been deposited to the ProteomeXchange Consortium through the PRIDE partner repository with the dataset identifier PXD058665. Publicly available sequence data used in this study with metadata and accession numbers are provided in Supplementary Table 9. Publicly available datasets were used in structural comparisons (PDB 5IR0 and PDB 2OB9). All other data supporting the

findings of this study are available in the paper and Supplementary Information. For gel and TEM image source data, see Supplementary Fig. 5. Source data are provided with this paper.

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

**Acknowledgements** We are especially thankful for the support of the icddr,b hospital and the Molecular Ecology and Metagenomics Laboratory staff. We thank A. Angermeyer for the initial bioinformatic investigation of PLE11 and members of the Seed laboratory for critical feedback and thoughtful discussion regarding this project. We thank the staff at the electron microscopy facility and the Vincent J. Coates Proteomics/Mass Spectrometry Laboratory Core Facility (RRID: SCR_025852) at the University of California Berkeley. We thank members of the Center for Structural Genomics of Infectious Diseases (CSGID) for determining the X-ray diffraction structure of the TAC from PLE1 (PDB 5IRO). CSGID was funded with US Federal funds from the National Institute of Allergy and Infectious Diseases, National Institutes of Health Department of Health and Human Services under contract number HHSN272201700060C. This work was supported by the National Institute of Allergy and Infectious Diseases (grant nos. R01AI127652 and R01AI153303 to K.D.S.). Its contents are solely the responsibility of the authors and do not necessarily represent the official views of the National Institute of Allergy and Infectious Diseases or NIH. icddr,b gratefully acknowledges the Government of the People's Republic of Bangladesh and the Global Affairs Canada for their unrestricted support.

**Author contributions** Y.M., C.M.B. and K.D.S. conceptualized the study. Resources (collection of cholera patient stool samples and isolation of bacteria) were contributed by M.T.I., M.S., T.A. and M.A., with project administration by M.S. and supervision conducted by T.A. and M.A. Isolation and purification of phages from stool was performed by J.E.F. Plaque assays were performed by Y.M., C.M.B. and J.E.F. Extraction of DNA for sequencing from clinical phage and bacterial isolates was performed by J.E.F. Genotype analysis was performed by J.E.F., Y.M., C.M.B. and K.D.S. M.M.M. performed bacterial phylogenetic analysis. qPCR was performed by J.E.F. Escaper phage generation was performed by J.E.F. and K.D.S. Construction of edited phages was performed by Y.M., C.M.B. and J.E.F. Construction of *V. cholerae* mutants was performed by Y.M., C.M.B. and J.E.F. Sample preparation for TEM was performed by Y.M., C.M.B. and J.E.F. and image collection and analysis was performed by Y.M. and C.M.B. Transduction assays were conducted by Y.M. and C.M.B. Western blotting, virion purification and mass spectrometry was performed by Y.M. and C.M.B. PLE genome alignments and bioinformatic analyses were performed by Y.M. and C.M.B. C.M.B. performed the phage genome phylogenetic analysis. VIRIDIC heatmaps were generated by Y.M. Y.M. performed TAC structural analysis. Y.M., C.M.B. and M.M.M. were responsible for visualization. Y.M., C.M.B. and K.D.S. wrote the original draft. All authors discussed the results, commented on and approved the final paper. K.D.S. supervised the project. M.A. and K.D.S. acquired funding.

**Competing interests** The authors declare no competing interests

**Additional information**
**Correspondence and requests for materials** should be addressed to Kimberley D. Seed.

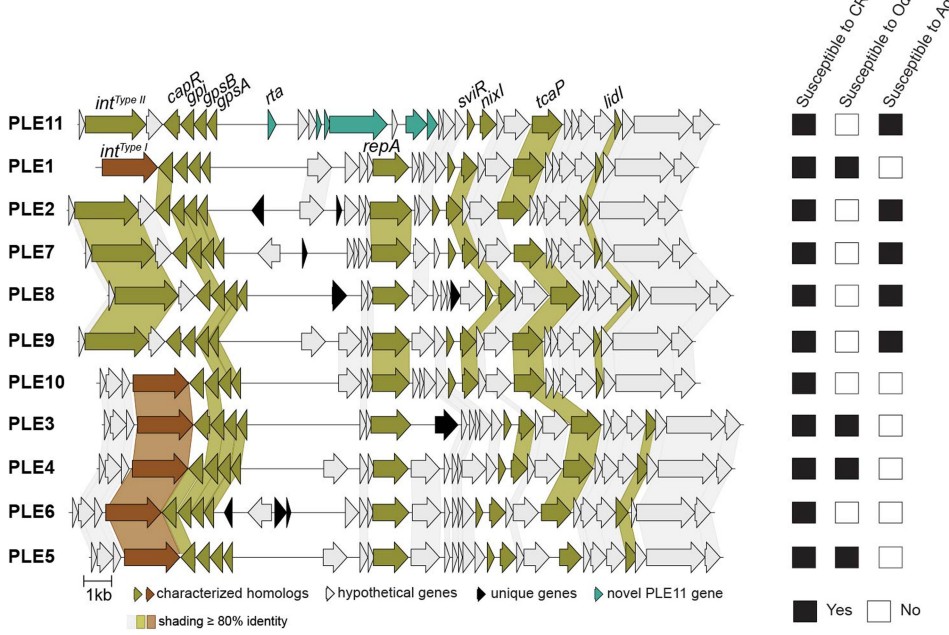

**Extended Data Fig. 1 | The new PLE variant, PLE11, inhibits ICP1 with known anti-PLE mechanisms.** Gene map comparison of *V. cholerae* PLEs. Genes were identified using tBLASTX and visualized using clinker. Similar genes encoding proteins in multiple PLEs have links drawn between them and are shaded if amino acid sequence identity >80%. Characterized genes from PLE1 and homologs in PLE11 are indicated; the Type II integrase was characterized in PLE2. The highly conserved regulatory RNA SviR (non-protein coding) is also shown. For each PLE, the expected susceptibility pattern to the known anti-PLE mechanisms is indicated (based on validated infection outcomes from characterized PLEs).

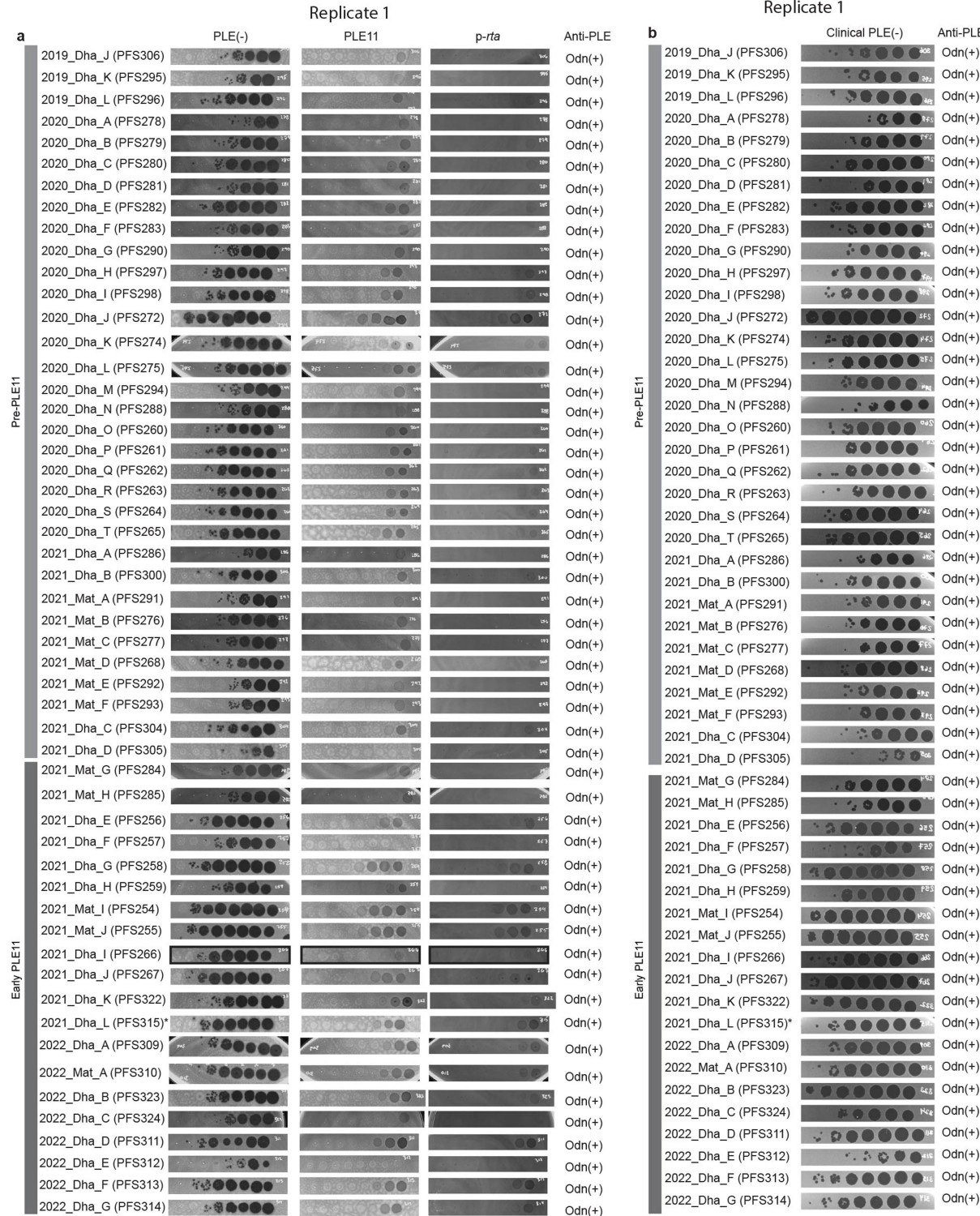

**Extended Data Fig. 2 | PLE11 protects *V. cholerae* from pre- and co-circulating Odn(+) phage isolates. a**) Plaquing of tenfold serially diluted ICP1 phage isolates from the pre-PLE11 and early-PLE11 periods (December 2019 to June 2022) on lawns of *V. cholerae* strain E7946, E7946 PLE11(+) and E7946 with a low-copy plasmid expressing PLE11 *rta*. Images used in the main text are indicated with black boxes. Biological replicates (n = 3) are shown in Supplementary Fig. 1a. **b**) Plaquing of tenfold serial diluted ICP1 phage isolates as in (a) on lawns of the clinical PLE(-) isolate BFS948. BFS948 was selected as the closest PLE(-) relative to the PLE11(+) isolate BFS783 used in Fig. 1. Biological replicates (n = 3) are shown in Supplementary Fig. 1b. For both panels, standardized ICP1 names are shown (denoting the year and location of isolation: Dha = Dhaka, Mat = Mathbaria, and lab designations (PFS#)). Shading indicates periods before (light gray) and the first year after (medium gray) PLE11 detection. All phage isolates from this period encode Odn as their sole anti-PLE mechanism; none of the whole genome sequenced isolates (n = 17) harbor substitutions in the TMP within the Rta-associated region (amino acid positions 314-387), but one isolate (marked by *) contains a TMP substitution (R132C) which does not provide resistance to PLE11 or Rta (Supplementary Table 2). For isolates genotyped by PCR for Odn vs CRISPR-Cas, the sequence of the *tmp* was not analyzed.

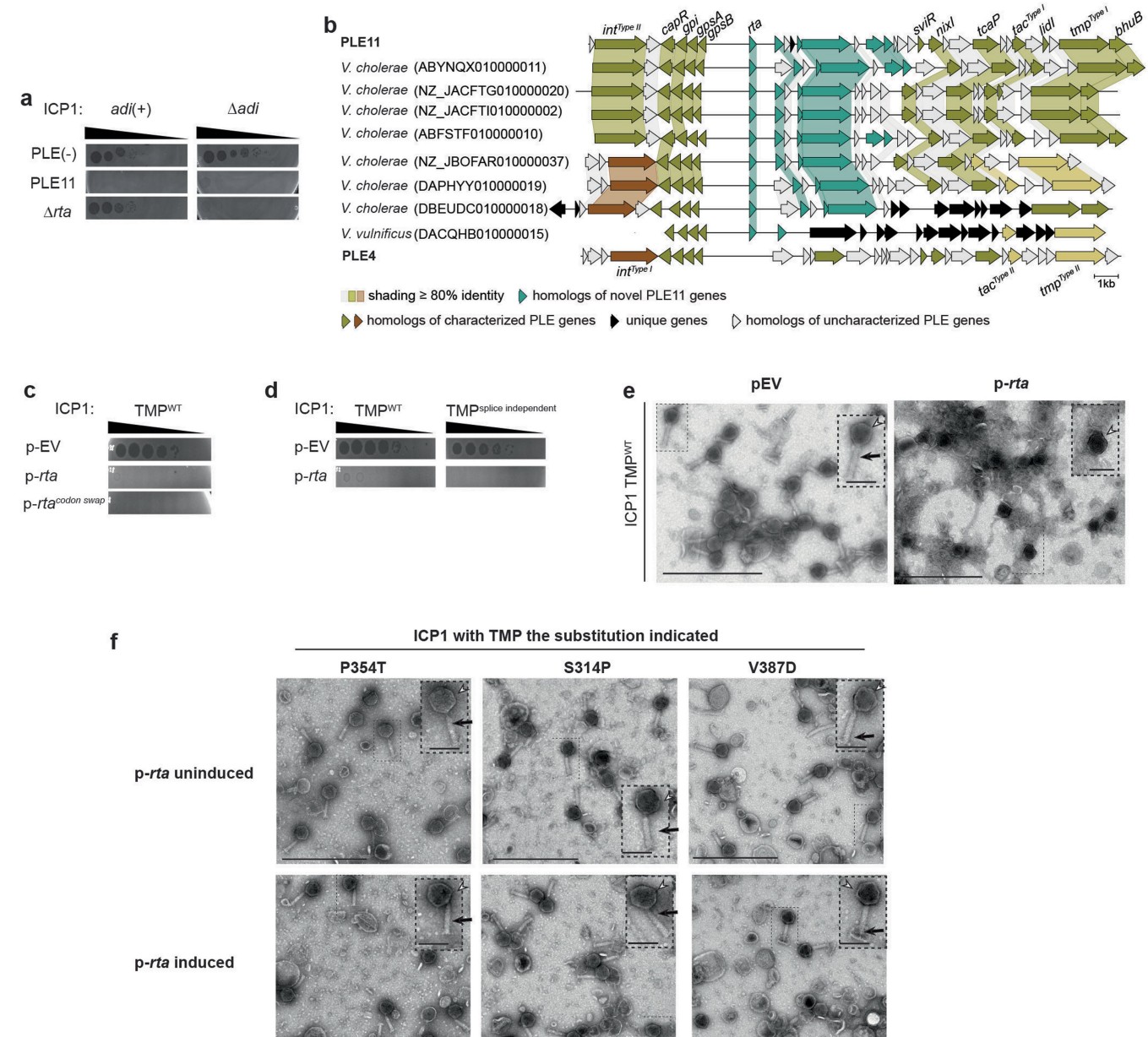

**Extended Data Fig. 3 | PLE11-encoded Rta restricts the assembly of tailed ICP1 virions. a**) Plaquing of tenfold serially diluted ICP1 Adi(+/-) on lawns of *V. cholerae* strain E7946 and its PLE11(+) and PLE11 mutant derivatives. The gray background is the bacterial lawn, and the dark spots are zones of killing. Biological replicates (n = 3) are shown in Supplementary Fig. 2a. **b**) Gene map comparison of genomic localization of Rta-homologs found in bacterial genomes. Rta homologs were identified using BLASTp and tBLASTX with PLE11 Rta as the query. The source nucleotide database, indicated in parentheses, was used to derive the gene neighborhood, which was then visualized using Clinker. Homology links for genes encoding proteins with amino acid sequence identity >80% are shaded. The PLE11 and PLE4 gene maps shown are references for characterized genes with type-1 and type-2 tail operons, respectively. Homologs of characterized PLE genes, uncharacterized PLE genes, and genes novel to PLE11 are indicated as in Extended Data Fig. 1a; unique genes defined as not found in any known PLE variant are shown in black arrows. The dashed line indicates the end of the contig. **c**) Plaquing of tenfold serially diluted ICP1

CR-Cas(-) on lawns of *V. cholerae* harboring an empty vector (pEV), a vector expressing Rta (p-*rta*), or a vector expressing codon-swapped *rta* (p-*rta^codon swap*). Biological replicates (n = 3) are shown in Supplementary Fig. 2b. **d**) Plaquing of tenfold serially diluted wild-type (WT) ICP1 CR-Cas(-) wherein production of the tape measure protein (TMP) involves splicing (TMP^WT) or a mutant constructed to bypass splicing (TMP^splice independent) on lawns of *V. cholerae* harboring pEV or p-*rta*. Biological replicates (n = 3) are shown in Supplementary Fig. 2c. **e, f**) Representative transmission electron micrographs (TEMs) of particles produced following (**e**) ICP1 infection of PLE(-) *V. cholerae* with pEV or p-*rta* with inducer (biological replicates (n = 3) are shown in Supplementary Fig. 2d) or (**f**) infection of *V. cholerae* with p-*rta* (+/-) inducer with ICP1 derivatives with relevant substitution in the TMP (n = 1). The arrowheads indicate DNA-filled capsids, and the arrow indicates a tail. The scale bars are 500 nm and 100 nm for the zoomed-out and insets, respectively. For TEM source data, see Supplementary Fig. 5.

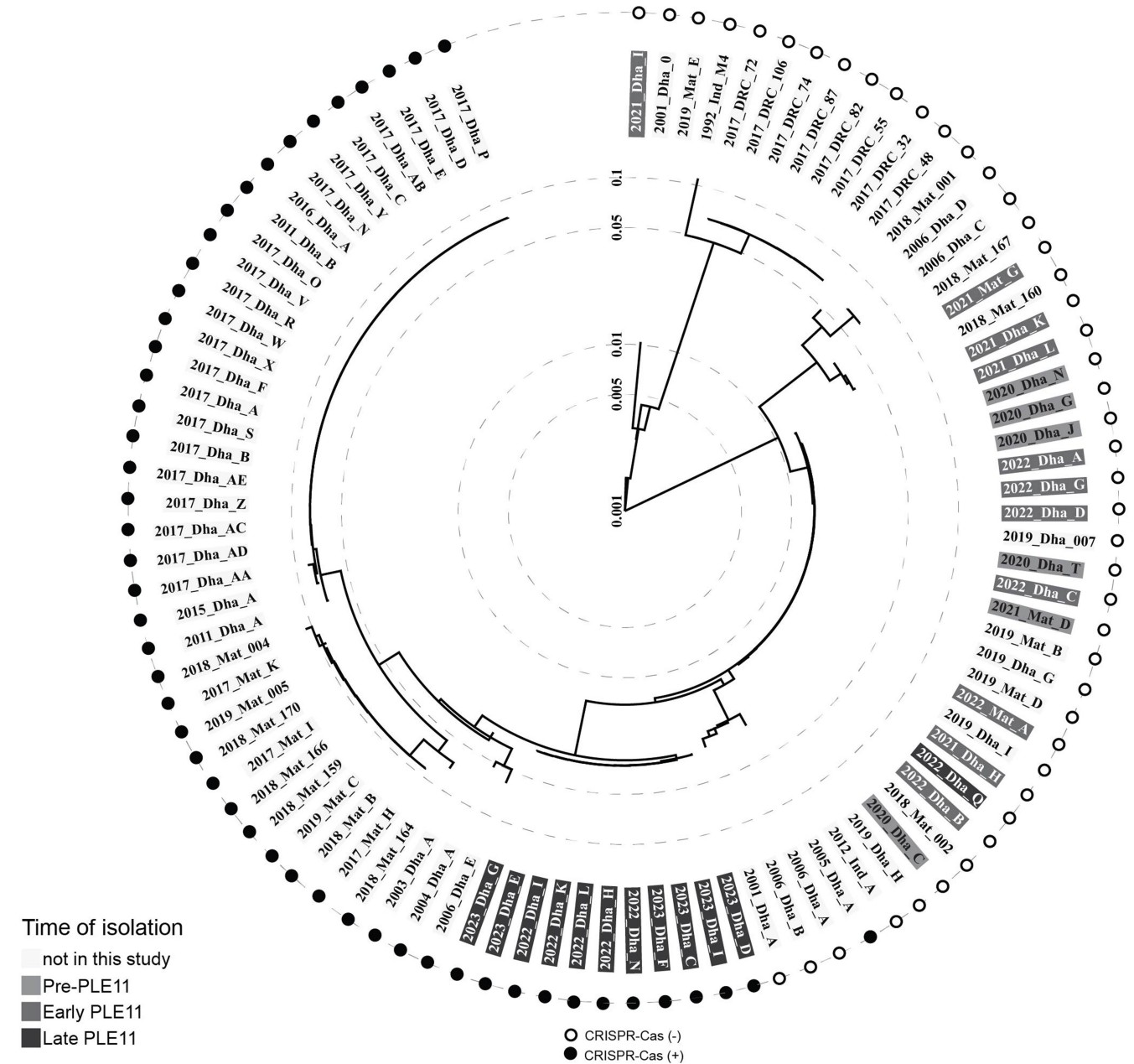

**Extended Data Fig. 4 | Phylogeny of all sequenced ICP1 isolates.** Phylogeny of 96 whole genome sequenced ICP1 phage isolates (29 from this study and 67 from previously published works[6] based on all translated open reading frames to determine overall similarity (using VipTree's tBLASTx-based algorithm). Filled and empty circles indicate the presence and absence of CRISPR-Cas, respectively. The gray shading indicates the period of isolation, as the legend indicates. Standardized ICP1 names denote year and isolation location (Dha=Dhaka, Bangladesh; Mat = Mathbaria, Bangladesh; Ind = India; DRC = Democratic Republic of Congo).

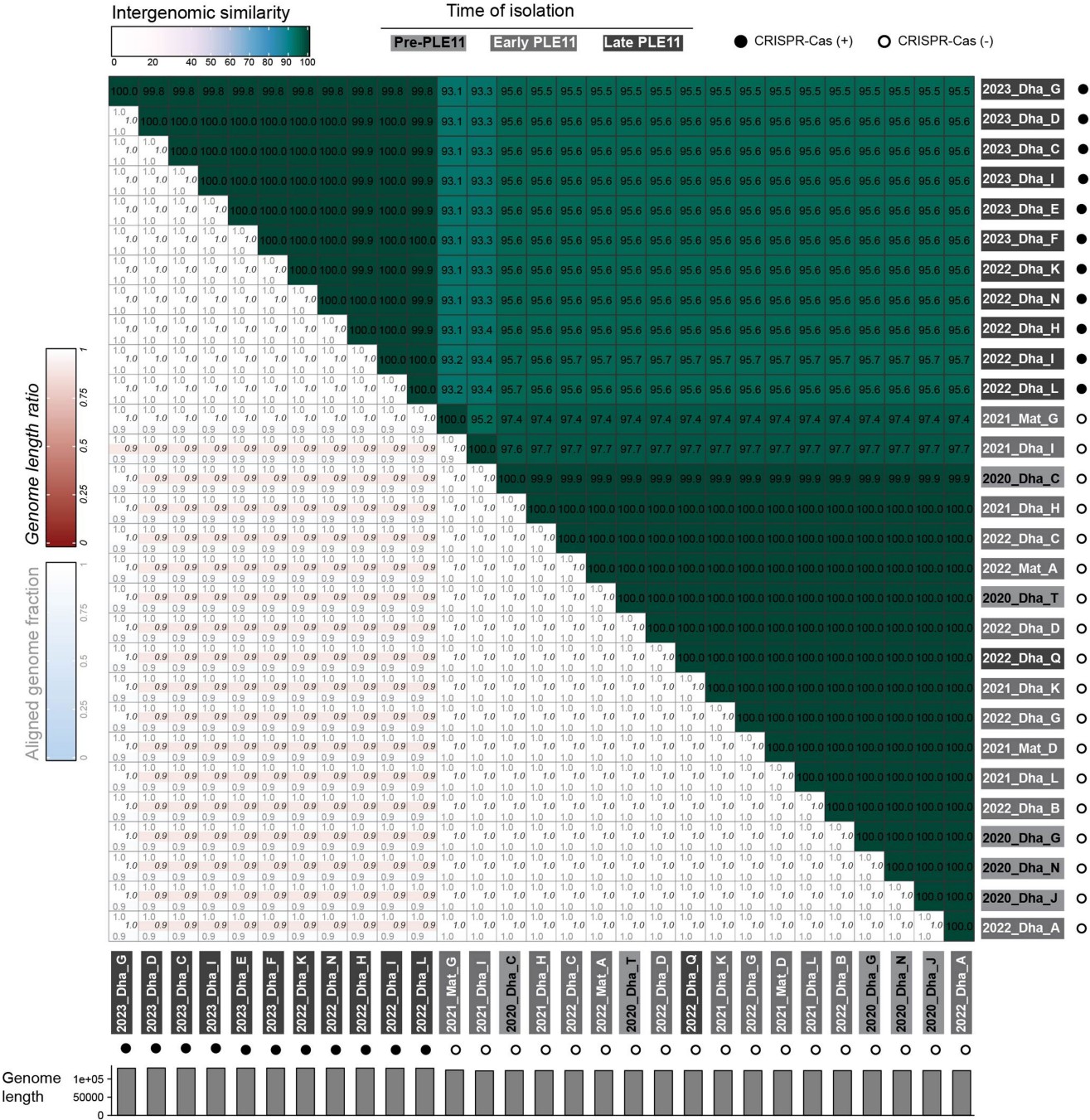

**Extended Data Fig. 5 | Intergenomic similarities between phages sequenced in this study.** VIRIDIC heatmap generated for the 29 ICP1 genomes sequenced during the period of this study. The intergenomic similarity values (right half) are displayed in a PurpleBlueGreen color scale (top), darker color signifies more closely related genomes, and the numbers represent similarity values rounded to a single decimal place as the default tool settings. The alignment indicators (left half) include the aligned genome fractions (light grey text, light blue gradient in the top and bottom of each square), corresponding to genomes listed on the row and column, respectively, and the genome length ratio (italic text, firebrick gradient in the middle of each square) for the two genomes in the pair. The lighter colors for aligned genome fraction represent that a larger fraction of genomes were aligned and lighter colors for the genome length ratio represent a smaller difference in the lengths of the genome pair. The absolute genome lengths of each phage are represented in the bar graph (bottom). Filled and empty circles next to phage names indicate the presence and absence of CRISPR-Cas, respectively. Phages lacking CRISPR-Cas encode Odn. The gray shading over the phage name indicates the period of isolation, as the legend indicates. Standardized ICP1 names denote year and isolation location (Dha = Dhaka, Bangladesh; Mat = Mathbaria, Bangladesh).

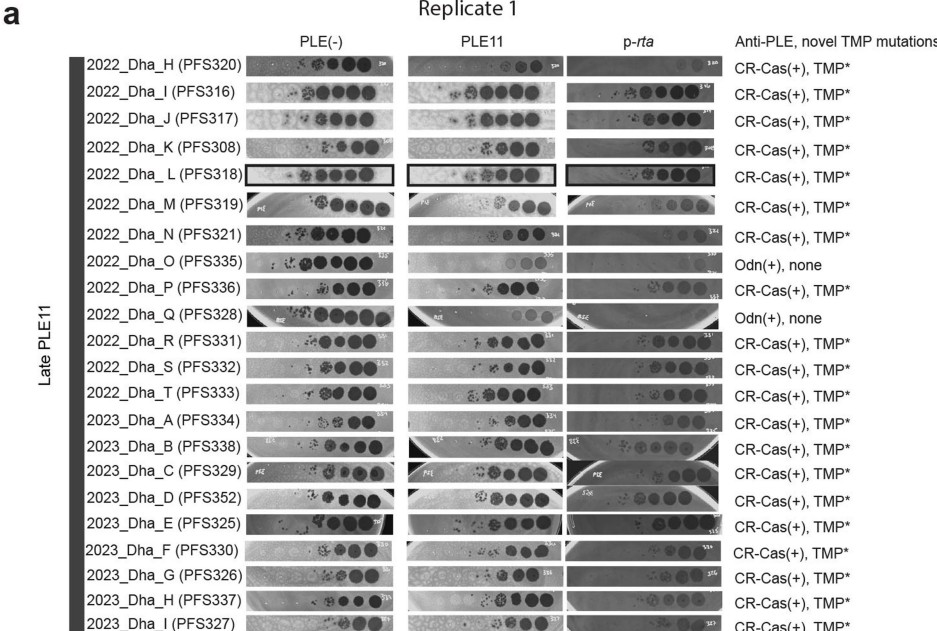

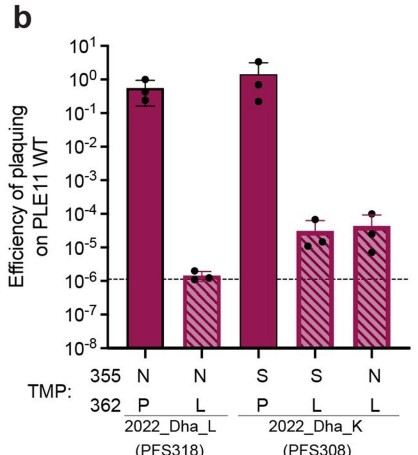

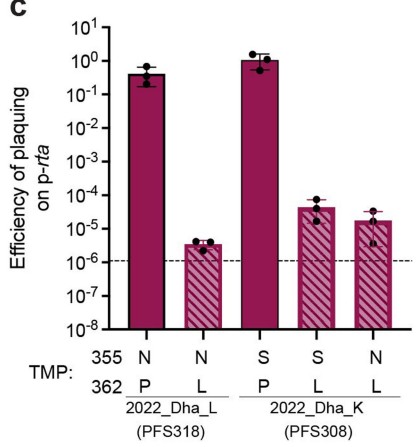

**Extended Data Fig. 6 | ICP1 isolates from the late-PLE11 period have evolved to overcome PLE11-mediated defense and Rta. a)** Plaquing of tenfold serially diluted ICP1 phage isolates from the late PLE11 period (July 2022 to December 2023) on lawns of *V. cholerae* strain E7946, E7946 PLE11(+) and E7946 with a low-copy plasmid expressing PLE11 *rta*. The gray background is the bacterial lawn, and the dark spots are zones of killing. Images that were used in the main text are indicated with black boxes. Standardized ICP1 names denote year and isolation location (Dha = Dhaka, Mat = Mathbaria, and lab designations (PFS#)). The dark gray shading indicates the period after PLE11 (late-PLE11 period, CRISPR-Cas(+) ICP1 co-circulation). Anti-PLE mechanisms and tape measure

protein (TMP) substitutions are indicated; TMP* refers to either substitution combination of L362P ± N355S; none indicates no substitutions. Biological replicates (n = 3) are included in Supplementary Fig. 3. **b** and **c)** Efficiency of plaquing on (b) PLE11(+) *V. cholerae* relative to a permissive PLE(-) strain or (c) on Rta-expressing *V. cholerae* relative to permissive *V. cholerae* harboring an empty vector, tested for ICP1 clinical isolates (solid bars): 2022_Dha_L (PFS318, isolate with L362P substitution) and 2022_Dha_K (PFS308, isolate with both N355S and L362P substitutions) and their respective engineered derivatives to revert the substitutions (hatched bars). Data are mean ± s.d. (n = 3 biological replicates); DL: detection limit.

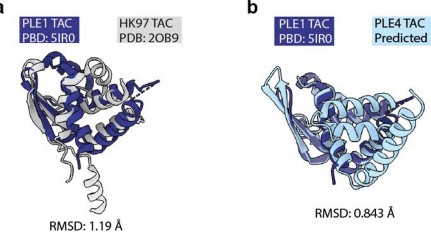

a

PLE1 TAC
PBD: 5IR0

HK97 TAC
PDB: 2OB9

RMSD: 1.19 Å

b

PLE1 TAC
PBD: 5IR0

PLE4 TAC
Predicted

RMSD: 0.843 Å

········ ········· Missing structure

**Extended Data Fig. 7 | PLEs encode a tail assembly chaperone structurally similar to a characterized tail assembly chaperone in phage HK97. a**) Smith-Waterman alignment between partially solved X-ray diffraction structures of the TAC from PLE1 (PDB:5IR0) and HK97 TAC (PDB: 2OB9), a characterized phage TAC. PLE11's putative TAC is 96.88% identical to PLE1's TAC. **b**) Smith-Waterman alignment between partially solved X-ray diffraction structures of the TAC from PLE1 (PDB:5IR0) and the predicted structure of the TAC from PLE4. The dotted lines (···) indicate the missing residues.

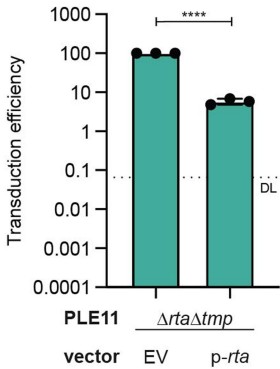

**Extended Data Fig. 8 | Expression of Rta *in trans* reduces transduction of PLE11 lacking its own TMP.** Transduction efficiency of Δ*rta*Δ*tmp* PLE11 with an empty (EV) or a plasmid expressing Rta relative to the EV control upon infection by ΔCRISPR-Cas ICP1. Data are mean ± s.d. (n = 3 biological replicates); DL: detection limit. ****p < 0.0001 (two-sided Student's t-test).

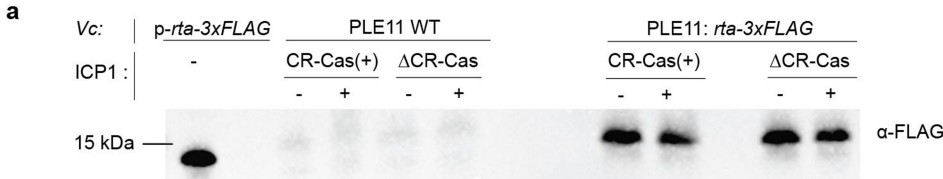

a

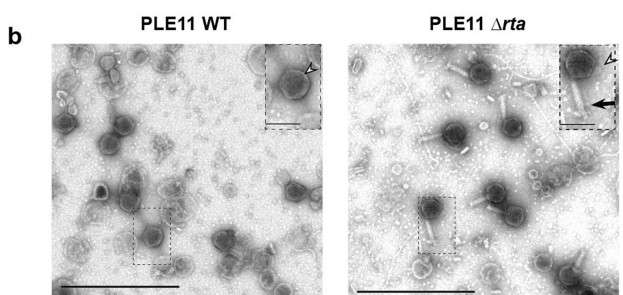

b

**PLE11 WT**　　　**PLE11 Δ*rta***

**Extended Data Fig. 9 | PLE11 Rta is expressed prior to ICP1 infection and restricts tail assembly of CRISPR-Cas(+) ICP1. a)** Western blot analysis of bacterial cell lysates collected pre-infection and post-infection by CRISPR-Cas(+/-) ICP1 on PLE11 *V. cholerae* encoding a 3x-FLAG tagged Rta in the native locus. PLE11 WT *V. cholerae* is used as a negative control, and PLE(-) *V. cholerae* expressing p-*rta-FLAG* is used as a positive control for the detection of Rta-3x-FLAG construct. For gel source data, see Supplementary Fig. 5. For detection of Rta-3x-FLAG in uninfected cells n = 2 biological replicates were included on the blot; detection of Rta-3x-FLAG post-infection by CR-Cas(+) and ΔCRISPR-Cas ICP1 was performed n = 1 each. **b)** Representative transmission electron micrographs (TEMs) of particles produced following CRISPR-Cas(+) ICP1 infection of PLE11 WT or Δ*rta V. cholerae*. The arrowheads indicate DNA-filled capsids, and the arrow indicates a tail. The scale bars are 500 nm and 100 nm for the zoomed-out and insets, respectively. Biological replicates (n = 3) are shown in Supplementary Fig. 4. For TEM source data, see Supplementary Fig. 5.

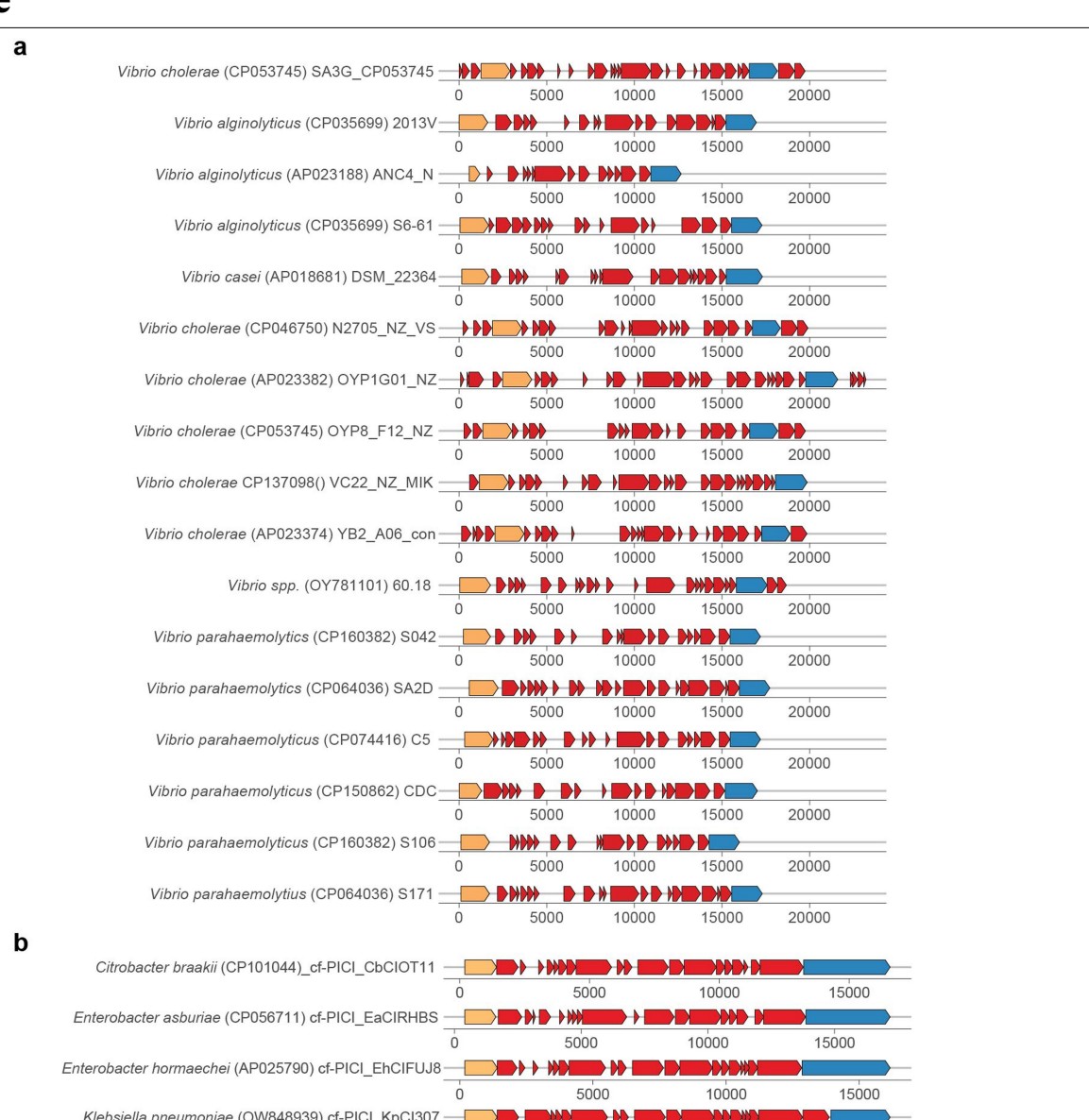

**Extended Data Fig. 10 | Putative *tmp* genes are encoded in diverse putative phage satellites. a**) Gene maps of PLE-like elements (identified in LeGault et al.[8]) These elements were queried for homologs of PLE11's tape measure protein (TMP) (using tBLASTn). **b**) Gene maps of a sampling of cf-PICIs (capsid-forming phage-inducible chromosomal islands, Alqurainy et al.[49]; a family of phage satellites unrelated to PLEs) encoding putative *tmp* genes. For a and b: Accession numbers are shown within the parentheses. Integrases and TMPs were predicted by homology to known or putative integrases and TMPs using HHPred. All other genes are indicated as arrows for simplicity.

# Reporting Summary

## Statistics

For all statistical analyses, confirm that the following items are present in the figure legend, table legend, main text, or Methods section.

| n/a | Confirmed | |
|---|---|---|
| ☐ | ☒ | The exact sample size (*n*) for each experimental group/condition, given as a discrete number and unit of measurement |
| ☐ | ☒ | A statement on whether measurements were taken from distinct samples or whether the same sample was measured repeatedly |
| ☐ | ☒ | The statistical test(s) used AND whether they are one- or two-sided<br>*Only common tests should be described solely by name; describe more complex techniques in the Methods section.* |
| ☒ | ☐ | A description of all covariates tested |
| ☒ | ☐ | A description of any assumptions or corrections, such as tests of normality and adjustment for multiple comparisons |
| ☐ | ☒ | A full description of the statistical parameters including central tendency (e.g. means) or other basic estimates (e.g. regression coefficient) AND variation (e.g. standard deviation) or associated estimates of uncertainty (e.g. confidence intervals) |
| ☐ | ☒ | For null hypothesis testing, the test statistic (e.g. *F*, *t*, *r*) with confidence intervals, effect sizes, degrees of freedom and *P* value noted<br>*Give P values as exact values whenever suitable.* |
| ☒ | ☐ | For Bayesian analysis, information on the choice of priors and Markov chain Monte Carlo settings |
| ☒ | ☐ | For hierarchical and complex designs, identification of the appropriate level for tests and full reporting of outcomes |
| ☒ | ☐ | Estimates of effect sizes (e.g. Cohen's *d*, Pearson's *r*), indicating how they were calculated |

*Our web collection on statistics for biologists contains articles on many of the points above.*

## Software and code

Policy information about availability of computer code

| Data collection | no software was used for data collection |
|---|---|
| Data analysis | Sequencing data was analyzed with fastp v0.23.2, Spades v3.15.4, snippy v4.6.0, IQ-TREE v2.2.0, ModelFinder, ViPTree version 4.0, tBLASTx version 2.17.0, VIRIDICweb, NCBI BLAST, and HHPred. Structural predictions were made using ColabFOLD on COSMIC2, and structural visualizations were done using UCSF ChimeraX. Genome visualizations were generated using Clinkerv0.0.28 and R 4.4., using the gggenes package. |

For manuscripts utilizing custom algorithms or software that are central to the research but not yet described in published literature, software must be made available to editors and reviewers. We strongly encourage code deposition in a community repository (e.g. GitHub). See the Nature Portfolio guidelines for submitting code & software for further information.

## Data

Policy information about availability of data

All manuscripts must include a data availability statement. This statement should provide the following information, where applicable:
- Accession codes, unique identifiers, or web links for publicly available datasets
- A description of any restrictions on data availability
- For clinical datasets or third party data, please ensure that the statement adheres to our policy

Sequence data for V. cholerae and ICP1 isolates from clinical samples and for ICP1 escape phages have been deposited in the NCBI Sequence Reads Archive (SRA)

under BioProject PRJNA1195958. The PLE11 sequence has been deposited to GenBank (accession PQ783903). The mass spectrometry proteomics data have been deposited to the ProteomeXchange Consortium via the PRIDE partner repository with the dataset identifier PXD058665 under the DOI 10.6019/PXD058665. Publicly available sequence data used in this study with metadata and accession numbers are provided in Supplementary Table 9. Publicly available datasets were used in structural comparisons (PDB:5IR0, PDB:2OB9). All other data supporting the findings of this study are available in the paper and Supplementary Information. For gel and TEM image source data, see Supplementary Fig. 5. Source data are provided with this paper.

# Research involving human participants, their data, or biological material

Policy information about studies with [human participants or human data](). See also policy information about [sex, gender (identity/presentation), and sexual orientation]() and [race, ethnicity and racism]().

| | |
|---|---|
| Reporting on sex and gender | this section is not relevant as the study did not focus on human cholera patients but on the bacterial and phage isolates from their stool; no human material was collected or studied |
| Reporting on race, ethnicity, or other socially relevant groupings | this section is not relevant as the study did not focus on human cholera patients but on the bacterial and phage isolates from their stool; no human material was collected or studied |
| Population characteristics | this section is not relevant as the study did not focus on human cholera patients but on the bacterial and phage isolates from their stool; no human material was collected or studied |
| Recruitment | Stool samples were collected from suspected cholera patients at the icddr,b Dhaka Hospital and the Government Health Complex in Mathbaria, Pirojpur, under protocol number PR-16083 approved by the icddr,b Ethical Review Committee, with written consent obtained from participants or their guardians. |
| Ethics oversight | Dr. Alam's protocol number PR-16083 is approved by the icddr,b Ethical Review Committee (ERC); Dr. Seed is not required to have an IRB as all samples have been de-identified |

Note that full information on the approval of the study protocol must also be provided in the manuscript.

# Field-specific reporting

Please select the one below that is the best fit for your research. If you are not sure, read the appropriate sections before making your selection.

☒ Life sciences      ☐ Behavioural & social sciences      ☐ Ecological, evolutionary & environmental sciences

For a reference copy of the document with all sections, see [nature.com/documents/nr-reporting-summary-flat.pdf]()

# Life sciences study design

All studies must disclose on these points even when the disclosure is negative.

| | |
|---|---|
| Sample size | Sample size was not calculated for comparative genomic studies because no standard or statistical approach for choosing the appropriate number of strains has been established. For all quantitative experiments and phenotypic observations using phage spot plates and electron microscopy, we chose to replicate experiments in biological triplicate as is routine to indicate reproducibility and allow for statistical analysis. Sample sizes for particle counting using EM images were determined by the density of particles on the grids generated from three biological replicates of particle purifications. TEMs of ICP1 with TMP substitutions were done one time each on three independent substitution mutants. The proteomics analyses of purified ICP1 and PLE11 particles were conducted once due to the high time and monetary costs, but results were corroborated through independent means (i.e. alterations to the structural composition of PLE particles obtained by an independent preparation of particles as assessed by western blot in Fig. 4d). Western blotting of Rta-3X FLAG was done one time which incorporated two biological replicates of the uninfected sample which was the key finding regarding Rta's expression prior to phage infection. |
| Data exclusions | No data were excluded from the analysis. |
| Replication | All reported findings were successfully reproduced through independent biological replicates or corroborating analyses as described above. |
| Randomization | No experimental groups or control groups were subjectively chosen therefore no randomization is required. |
| Blinding | Blinding is not relevant to the experimental design as all data were obtained objectively. |

# Reporting for specific materials, systems and methods

We require information from authors about some types of materials, experimental systems and methods used in many studies. Here, indicate whether each material, system or method listed is relevant to your study. If you are not sure if a list item applies to your research, read the appropriate section before selecting a response.

## Materials & experimental systems

| n/a | Involved in the study |
|-----|----------------------|
| ☐ | ☒ Antibodies |
| ☒ | ☐ Eukaryotic cell lines |
| ☒ | ☐ Palaeontology and archaeology |
| ☒ | ☐ Animals and other organisms |
| ☒ | ☐ Clinical data |
| ☒ | ☐ Dual use research of concern |
| ☒ | ☐ Plants |

## Methods

| n/a | Involved in the study |
|-----|----------------------|
| ☒ | ☐ ChIP-seq |
| ☒ | ☐ Flow cytometry |
| ☒ | ☐ MRI-based neuroimaging |

## Antibodies

| | |
|---|---|
| Antibodies used | Custom antibodies generated in rabbits by GenScript were used for anti-ICP1 baseplate hub (BhuB) (PolyExpress Premium Cat. no. SC1676), anti-ICP1 tape measure protein (TMP) (PolyExpress Premium Cat. no. SC1676), anti-PLE11 tape measure protein (TMP) (PolyExpress Premium Cat. no. SC1676) and anti-ICP1 capsid (PolyExpress Gold cat. no. SC1649). Anti-FLAG antibody was purchased through GenScript (cat. No. A00170-40). |
| Validation | Custom Antibodies were validated by the manufacturer using ELISA and western blot analyses of purified antigen. Anti-FLAG: validated by "Western blot analysis of DYKDDDDK tagged fusion proteins expressed in E. coli cell lysate" Additionally, we included positive and negative controls in our western blotting for validation of the detection of FLAG tagged RTA (in ED Figure 9, positive control was plasmid expressing rta 3xFLAG and untagged PLE11 strain was used as negative control). |

## Plants

| | |
|---|---|
| Seed stocks | n/a |
| Novel plant genotypes | n/a |
| Authentication | n/a |

