## [Peer Review File · Nature]

Capturing dynamic phage-pathogen coevolution by clinical surveillance

Corresponding Author: Professor Kimberley Seed

Version 0:

Reviewer comments:

Referee #1

(Remarks to the Author)

This is a very interesting manuscript by Mathur and Boyd et. al. studying the evolution of *Vibrio cholerae* and its predominant ICP-1 phage both in real-world clinical samples and laboratory conditions. The authors make several impactful findings, and this manuscript was enjoyable to read given all the different perspectives in which they addressed this question. By surveying *V. cholerae* clinical isolates and ICP-1 phage in Bangladesh, the authors first found that a new *V. cholerae* strain emerged which encoded a novel PLE they named PLE11. They demonstrated that PLE11 provided protection from all contemporary ICP-1 phage. However, after around 9 months ICP-1 evolved variants that were able to overcome PLE11. The authors then studied this interaction in the lab showing that PLE11 encodes a protein named Rta that presumably inhibits the tape measure protein (TMP) of ICP-1, causing ICP-1 virions to be tailless. Mutations to a specific region of the protein both isolated in the lab and observed in clinical ICP-1 isolates overcome Rta, presumably as they no longer interact. The authors then show that PLE11 overcomes its own Rta by encoding its own TMP, which is resistant to it. This is the first demonstration of a satellite phage defense that inhibits tail assembly. It is also striking to see the natural evolution of this interaction play out in the real world. The manuscript was very well written and the data were strong. The weakest aspect of the manuscript is a lack of demonstration of Rta interaction with TMP, either from ICP-1 or PLE11. I also have several minor comments/questions about the study. Overall, I think this is an excellent, high-impact study that has many important findings which will be valuable to the field.

1. It is interesting that the majority of isolated strains rapidly became PLE11+, but for some time all the ICP-1 phages that were isolated were unable to replicate in these isolates. This leads me to wonder about the hosts for ICP-1 during this time. What is the small percentage of PLE11- strains or some other unidentified host? Also, do the authors have any data that speak to the prevalence of ICP-1 during this time period. One prediction would be an overall reduction of observable ICP-1 given the host resistance.
2. Following this point, Fig. 3a seems to indicate that indeed phage numbers decreased as the PLE11+ strains emerged in 2022, but is this quantitative of significant? Can the authors make that claim with the data that they have?
3. Lines 251-253-It is unclear what the evidence for the emergence of a new CRISPR-Cas+ ICP-1 phage versus replacement of the Odn gene. Can the authors clarify?
4. Fig. 3e-Can the N355S mutation confer resistance to Rta or is resistance mediated solely by L362P? The N355S mutation was not tested on its own. It is curious that there are two mutations in the tape measure protein if they both confer the same resistance to Rta. This is total speculation, but if N355S does not confer resistance to the Rta of PLE11, perhaps it confers resistance to a different Rta that these phages have encountered.
5. If a PLE11+ host was infected by a CRISPR-Cas ICP-1 phage, then the Rta from PLE11 would restrict ICP-1 but the CRISPR-Cas would restrict propagation of the PLE11. The authors hypothesize that this mechanism allows vertical transmission of the PLE11 and thus can be selected for. This is an interesting idea, and I think it could be better noted in Fig. 4h to emphasize that in this case the PLE11 is maintained in the genome. Along this line, would the genome copy of PLE11 be targeted by the ICP-1 CRISPR-Cas array?
6. The Rta protein would be very useful for the host even in the absence of the PLE. Have the authors searched *V. cholerae* genomes (or more broadly) to see if the bacterial have acquired Rta-like proteins as a mechanism of phage defense? Given its small size it may have been overlooked.

7. The authors could use AlphaFold to try and model the interaction between Rta and the ICP-1 TMP (and PLE11 TMP). It would be interesting to see if the region they identified resistance mutants maps to a binding domain.
8. One piece of data that is lacking is demonstrating the interaction of Rta with the ICP-1 TMP. The authors could try two-hybrid type or co-IP approaches to demonstrate interaction of Rta with the WT TMP but not the resistant mutant TMPs. Such experiments could be done with the PLE11 TMP too (presumably no interaction).

Referee #3

(Remarks to the Author)

This manuscript by Mathur et al. is a tour de force, expertly combining clinical surveillance with detailed phenotypic characterizations, and especially a lovely parallel of in-lab experimental evolution mimicking the environmental evolution. I loved the core conceit, am intrigued by the mechanism, and find that the claims about mechanism of rta, evolutionary pressure on ICP and PLE11 are all extremely well-supported, often with entirely orthologous (or at least independent) approaches.

As an expert in phage biology, I feel qualified to comment on every aspect of the manuscript.

Narratively, the flow of information was logical and the experiments well justified through the results. However, the manuscript did struggle a little – especially in the abstract and discussion – to balance the ‘new anti-phage mechanism’ and ‘clinical surveillance/evolution’ angles of the paper. The gap in knowledge presented in the introduction is showing that (and how) phage predation drives pathogen evolution in disease contexts. For the “that” aspect of this claim, it is almost certainly true, probably largely considered true by all phage biologists, and is a little bit peripheral to the core findings of the paper (see moderate concerns) – although in this I feel I’m probably being more of a devil’s advocate than need be, given the claim is not that extraordinary, and my objections are pretty nit-picky. The ‘how’ aspect is stronger in the paper, although again falls a tiny bit short of being bullet proof. I know this sounds damning, but to me this is only a narrative concern – the findings of this paper *really* excite me as a truly wonderful pairing of surveillance and experimental data, as a new mechanism of phage resistance that raises some extremely interesting ecological questions about ‘altruism’ beyond those of traditional ABIs, etc.

I also felt the title itself sold the work short – making no mention of the experimental work or its conclusions.

Critical: No critical flaws that require a rethinking or challenging of the premise.

Major:

- There is only one experiment where I felt some experiments had been omitted – Fig 3d. I think the phage TMP mutants should be shown for contrast, as should the phages on a ple11 rather than just the pRTA? I am almost certain these experiments will show nothing (unless the transduction efficiencies – see minor comments below – are actually further apart from the wt system than is suggested by that presentation), but they are quick, easy, and potentially important.

Moderate:

- I found it somewhat challenging to assess the number of ‘independent’ strains at a number of junctions. This goes for both phages and *V. cholerae*. For instance, while the 53 strains (Fig 1c) are presumably all from different stool samples, it’s not clear to me how clonal they are, or what level of diversity they are. Whether they are 100% clonal, or all slightly different isn’t fatal to any of the conclusions drawn, but I feel the current narration does obfuscate the level of independence of the replicates. I found this consistent throughout (e.g. L81, 88 (“91%”), 98 (“all co-circulating phage isolates”). Similarly “all escape mutants” (L181), or “one or more aa substitutions in the TMP” (L255) isn’t really quantifying the number of independent events, especially since ‘largely clonal’ isn’t described (L253).

- The nit-pick I alluded to about the narrative: I think “demonstrates” is too strong a word for L103, and in the discussion. The fact that Ple11 confers resistance is demonstrated. The fact that ICP1 evolution is driven by PLE11 is twice demonstrated; in vitro and in surveillance. The fact that *Vibrio* evolution is driven by ICP1 and that PLE11 ‘conferred a selective advantage to *V. cholerae* in situ’... is a claim that stretches just a little bit further. Similarly “propound fitness advantage” (L236) conferred to *Vibrio* isn’t... directly supported, just inferred. As a phage biologist who sees everything bacteria do as phage-driven, I am ready to believe this claim, especially given the associations between ICP1 and lessened disease states. But I think there is an alternative explanation that is possible – that all of this just reflects an arms race between PLE11 and ICP1, and reflects only selection on these. It may be that *Vibrio*’s fitness is drastically affected by the sensitivity to ICP1, yes, or it may be that *Vibrio*’s fitness is affected by PLE11 even in the absence of ICP1 (less likely, but possible), or it may be that this lineage was fitter for other reasons, happened to co-associate with PLE11, which happened to put selective pressure on ICP1, but that the reason for the better fitness in the current environment has more to do with the lineage than the PLE11 locus.

Minor:

- The claims that “evidence of phage-driven selection of ... *V. cholerae* has not been demonstrated” is ... I understand the authors, but given the ctx-phage association, I’m not sure it’s true.

- Can the primers for Fig 1 F not detect degradation directly (e.g. amplify the CRISPR-targeted site?). This evidence is compelling that replication is interfered with, and I agree that the most likely explanation is – given the CRISPR system – nucleolytic, but it seems odd not to show cleavage.

-The LOD is fairly low (for phage work! Obviously) in your assays in general – but especially in F3 A and C. I think I get why C is – presumably these are stocks around 10^6 pfu/ml plated on the sensitive hosts, but expressed RTA is harder to bypass. Might be nice to show it on an empty control (pEV) as your baseline EOP of 1, rather than the evolved mutants as the EOP of 1? This would allow us to assess the relative change in sensitivity across the two panels.

-Unless I am mis-remembering, Mini/Mind-Flayers (Mini is the satellite) does not hijack tails for virion assembly (L322).

- I quibble a little with the claim in L322 that this is not in keeping with the paradigm – almost all the information to date is compatible with it. Capsid-modifying proteins denying the capsid to the helper phage is very similar (conceptually) to this, the difference is – as is later pointed out – that in these interferences with the helper phage, no functional satellite particles are released.

- “Our results indicate that all PLEs encode Rta-independent mechanisms to disrupt phage tail assembly”. I am not sure where this claim is coming from – presumably supported by the data about all the tested PLEs requiring ICP tail proteins, but I don’t think the authors demonstrated that hijacking these is ‘disrupting tail assembly of ICP. At least in this paper, only Rta is demonstrated to interfere with assembly.

- There’s a minor narrative point about altruism that I think is confounded by some of the writing – and I think may actually be a much bigger ‘deal’ than the authors make it out to be. In L456, the implication is that this is similar to other uses of the word altruism in phage defence (e.g. ABIs, even when encoded on a prophage, etc). As the authors correctly note, this is actually quite different, because by stopping phage spread through the population PLE11 is actively interfering with its own spread through the population, so it’s not just sacrificing one copy of a PLE11 (presumably the neighbouring cell has one too – as do all the nearby ones), as is the case in most of these so-called altruistic systems, it’s also interfering with the mechanism of spread of all its con-specific neighbours by preventing them from experiencing a productive helper phage infection. If prophage encoded rexAB “wins” by eradicating all T4-infected E. coli, what it leaves is plenty of E.coli lambda lysogens, and maybe even some sensitive E. coli for lambda to eat later. If rta “wins” by preventing the driving ICP1 extinct, PLE11 loses all ability to be horizontally transmitted.

Line-by-Line:

25: “selection of ICP1 that achieves a convergent evolutionary outcome” wasn’t clear this was within the patient population.

26-28: Felt like a bit of a non-sequitur.

40: What does “predominantly lytic” mean?

114: This suggests an ABI mechanism, but the language is a bit jarring – and would be especially so to a non-expert. The distinction between protecting from plaquing and protecting the bacterial genome integrity matters.

L248: “the anti-PLE mechanisms” should read “the known anti-PLE mechanisms”

L372: “was restored significantly” provide magnitude and statistical test? Similarly, for ‘markedly below’ (L377). I actually think the ~ 1 log decrease from the delta TMP is smaller than I’d expect – implying either that only 90% of particles are satellite TMPs, or that in its absence, ICP1 TMP is still at least 10% ‘as good’. That’s not too shabby!

L389 should reference the figure (4g?)

L463 might be a good place for a paragraph break.

L493-494: This sentence felt awkward, as it seemed to imply that the divergence of this particular system is striking, relative to the hundreds of known systems. I think maybe the word ‘from’ is what threw me off, as I believe the intention was to highlight that there is an incredible diversity of systems discovered to date, but that the idea (as elaborated on the next line) of tail-interfering is relatively new and interesting.

As is often the case when a manuscript excites me - I have some points of idle curiosity I wanted to raise, none of which are germane to the paper. The authors should not feel any need to address these, they simply are an a testament to my engagement and interest in the work:

- If the odn-sensitivity region were added to PLE11, would this allow RTA-sensitive but Odn+ ICP to replicate?

- If a CRISPR system targeted the rta gene, would this allow otherwise RTA-sensitive ICP to plaque?

- I understand there’s no need to test the natural N355S mutation by itself (since it never occurred on its own), I’m just curious if it has a phenotype.

- The coding density around rta is suspiciously low for a phage/phage-like element. What’s going on there?

- Signed Review: Alexander Hynes

Referee #4

(Remarks to the Author)

A. Mathur et al convincingly provides a link between the dynamics of ICP-1 infection, the defence systems deployed against it, and the clinical consequences in the form of Cholera dynamics. The paper provides mechanistic insights into how PLE11 sidestepped ICP-1’s adaptations to handle PLE1, and documents ongoing co-evolution between the phage and the satellite.

B. These mechanistic insights include the description of a novel defense pathway down to its molecular components, while also describing the implications of this evolutionary path on the epidemiological trends observed. It also suggests possible routes for future phage adaptation and makes a convincing argument that surveillance should be expanded to consider the phage and anticipate future sweeps mediated by negative frequency dependent selection (and raising the fascinating

question of how low frequency diversity is maintained in the population).

C. The surveillance data are convincing in demonstrating the sweep studied in the MS, and the ongoing evolution of the elements involved. The experimental results strongly support the hypothesized mechanism driving the evolutionary and epidemic dynamics.

D. Both are appropriate

E. some of these conclusions appear to contrast with the contents of the companion paper, and illustrate the limits of our ability to robustly identify the causes of the observed dynamics. The presence of ICP1 is stated to correlate with less severe disease, while Barton et al states that the presence of PLE1 defenses also do so. Is this because of the phage or the defenses against it? Additionally, the discussion states “Bangladesh is part of a wider transmission network”, but that is in tension with the findings of Barton et al that for much of their study, Bangladesh was not. It might be worth addressing how generalizable and clinically relevant the PLE11 findings are, given the lack of transmissibility implied elsewhere. The evidence that PLE11 helped the bacteria defend against resident ICP1 is solid, even more so with the co-evolution evidence. However the claim of PLE11 is a *main* contributing factor of the outbreak itself is less secure. Could the pathogenicity resulting in the outbreak be due to some other genetic difference on which PLE11 piggybacked because it indeed conferred resistance to the phage? The fact that there were 2 separate PLE11 introductions is supportive if not definitive, but perhaps giving a more detailed breakdown of the other background genetic differences between the outbreak and non-outbreak related strains would be helpful? The results should also be discussed in the context of the data in the companion MS which suggests lower pathogenicity for PLE11 strains (fig 3a).

F. In the laboratory evolution experiments, ICP1 CRISPR-Cas(+) mutants are evolved to escape PLE11 and find TMP mutations comparable to those seen in the population. Nevertheless, when comparing to the observed environmental ICP1 evolution “...our surveillance revealed the predicted shift from phages with the ineffective anti-PLE mechanism Odn to phages with CRISPR-Cas that overcomes PLE11’s conserved anti-phage defenses and substitutions in the TMP to counter PLE11’s unique Rta-mediated phage defense. While experimental evolution studies could not select for CRISPR-Cas(+) isolates from an Odn(+) population— due to the lack of pre-existing variation in genetically homogenous stocks—natural populations must maintain some level of genetic diversity, enabling selection of rare alleles and oscillations of the dominant genotypes.” Is the idea that some ICP1 strain with both Odn(+) and CRISPR-Cas(+) evolved to lose Odn, or that a low prevalence CRISPR-Cas(+) strain gained an advantage and swept through the population? Clarification of this would be helpful. Additionally, would it be possible to perform an evolution experiment with a more diverse population to validate these claims?

In addition in Figure 1c, an arrow marks the first isolate harboring PLE11 within the phylogenetic tree, which is indicated to be in the Early PLE11 period. However, some isolates in the same clade appear to be PLE11(+) and found in the pre-PLE11 period. What does this mean?

The association of PLE11 with a large outbreak is important, but the companion paper also asserts that PLE11 is associated with less severe disease. This is not necessarily wholly contradictory, but deserved to be addressed.

G. the references are appropriate, but is relevant to a long-standing question in cholera epidemiology, as to whether disease dynamics reflect environmental factors or the evolution of the bacteria. In this case, it appears to be the evolution of the bacteria. Although the question of why phage are apparently important in some locations but not others may be a matter of ecology. It could be worthwhile citing some of this background for context.

H. This is an extremely well written and presented paper throughout. Congratulations are in order.

Referee #5

(Remarks to the Author)

I co-reviewed this manuscript with one of the reviewers who provided the listed reports.

Version 1:

Reviewer comments:

Referee #1

(Remarks to the Author)

This manuscript did a great job addressing the reviewers’ comments. Well done!

Referee #3

(Remarks to the Author)

I was quite favourable to the initial manuscript, and the authors have been thorough in their rebuttal. I am more than satisfied with the final product.

On my single major concern at the time was asking for them to show the TMP mutants EMs, and while I might quibble as to

their placement in supplemental, I think the results themselves (expected by both myself and the authors) do a good job of crossing the proverbial Ts. The omission of the PLE versions of this is well justified by the authors.

Other reviewers raised concerns about demonstrating the direct rta-TMP interaction biochemically, but I did not feel this was lacking from the manuscript. The new efforts with a codon-shuffled, and an orthologue are nice additions, however.

The new information provided in addressing other reviewer concerns that establishes the pre-expression of RTA and its effectiveness even against anti PLE phages is a very nice addition, and also addressed one of my "idle thoughts" at the end of my review.

For the title, maybe "Validating dynamic phage-pathogen coevolution captured by clinical surveillance". Demonstrating? I don't feel strongly about it! Both are more than 6 characters. Maybe the editors will allow you leeway!

The point about the varying LOD is understandable, if a little suboptimal (it could lead the reader to think the effects are very different in magnitude in Fig 2A, C), and in my case, led to an assumption that pRTA was harder to bypass for the wt than PLE11 (which the authors note is entirely refuted by Fig 3E/F). This presentation, while suboptimal, is not misleading and is sufficient to support the claims in the manuscript.

Minor points:

Is the main body missing a header (after the summary?).

Referee #4

(Remarks to the Author)

Thank you for the opportunity to look at this paper and see how the authors responded. One of the other reviewers described the work as "a tour de force", which was actually a term that sprang to mind for me reviewing the previous version. The updates to the MS make it even stronger.

I'd like to thank the authors too for their thoughtful examination of the supplemental data from the accompanying MS. I agree with their comment about the value of multilayered surveillance.

Referee #5

(Remarks to the Author)

I co-reviewed this manuscript with one of the reviewers who provided the listed reports.

Referee #1

This is a very interesting manuscript by Mathur and Boyd et. al. studying the evolution of *Vibrio cholerae* and its predominant ICP-1 phage both in real-world clinical samples and laboratory conditions. The authors make several impactful findings, and this manuscript was enjoyable to read given all the different perspectives in which they addressed this question. By surveying *V. cholerae* clinical isolates and ICP-1 phage in Bangladesh, the authors first found that a new *V. cholerae* strain emerged which encoded a novel PLE they named PLE11. They demonstrated that PLE11 provided protection from all contemporary ICP-1 phage. However, after around 9 months ICP-1 evolved variants that were able to overcome PLE11. The authors then studied this interaction in the lab showing that PLE11 encodes a protein named Rta that presumably inhibits the tape measure protein (TMP) of ICP-1, causing ICP-1 virions to be tailless. Mutations to a specific region of the protein both isolated in the lab and observed in clinical ICP-1 isolates, overcome Rta, presumably as they no longer interact. The authors then show that PLE11 overcomes its own Rta by encoding its own TMP, which is resistant to it. This is the first demonstration of a satellite phage defense that inhibits tail assembly. It is also striking to see the natural evolution of this interaction play out in the real world. The manuscript was very well written and the data were strong. The weakest aspect of the manuscript is a lack of demonstration of Rta interaction with TMP, either from ICP-1 or PLE11. I also have several minor comments/questions about the study. Overall, I think this is an excellent, high-impact study that has many important findings which will be valuable to the field.

We thank the reviewer for their kind words and refer them to the detailed response to the critique concerning the interaction of Rta with the TMP in point 8 below.

1. It is interesting that the majority of isolated strains rapidly became PLE11+, but for some time all the ICP-1 phages that were isolated were unable to replicate in these isolates. This leads me to wonder about the hosts for ICP-1 during this time. What is the small percentage of PLE11- strains or some other unidentified host? Also, do the authors have any data that speak to the prevalence of ICP-1 during this time period. One prediction would be an overall reduction of observable ICP-1 given the host resistance.

We thank the reviewer for these insightful questions.

Regarding the hosts for ICP1 during the early emergence of PLE11, we anticipate that the remaining PLE11-negative subset of the *V. cholerae* population served as the primary hosts. This is supported by our finding (as noted in the manuscript) that Odn(+) phages were exclusively recovered from stool samples containing PLE-negative *V. cholerae*. While environmental strains of *V. cholerae* could also serve as hosts, we did not collect environmental samples and therefore cannot comment on their role in ICP1 dynamics.

We agree that a reduction in ICP1 abundance would be expected as the susceptible host population (PLE11-negative strains) decreased. While we cannot quantitatively track this change due to the inherent challenges of quantifying phage abundance in stool samples, we can provide the following qualitative information for the reviewer's context. Before the detection of PLE11, approximately 53% of the stool samples we surveyed were positive for phage. This percentage decreased to 16% during the early PLE11 period. This observation is consistent with the reviewer's prediction of a reduction in the prevalence of ICP1. However, given the limitations in quantitatively measuring phage in clinical samples, we do not feel it is appropriate to include these specific numbers in the final manuscript.

2. Following this point, Fig. 3a seems to indicate that indeed phage numbers decreased as the PLE11+ strains emerged in 2022, but is this quantitative of significant? Can the authors make that claim with the data that they have?

As we stated in our previous response, we do not feel we can confidently claim a statistically significant decrease in phage numbers. The methods used for sample collection, storage, and processing, while suitable for detection of phages, are subject to inherent variability that prevents quantitative analysis. For example, heterogeneity in stool sample consistency and a lack of standardized preservation times directly impact our ability to accurately quantify phage abundance per unit volume. The samples were also not processed on site for quantitative PCR or metagenomic analyses, which would be required for such analyses.

3. Lines 251-253-It is unclear what the evidence for the emergence of a new CRISPR-Cas+ ICP-1 phage versus replacement of the Odn gene. Can the authors clarify?

We were unable to substantiate this claim with additional formal evidence beyond the phylogenetic analyses presented. A recombination event that resulted in CRISPR-Cas replacing Odn could not be ruled out, hence we have removed this statement and the text now reads as follows:

This genotype shift reflected a replacement of the predominantly clonal Odn(+) ICP1 population with phylogenetically distinct, largely clonal CRISPR-Cas(+) ICP1 isolates (Fig. 3c, Extended Data Fig. 4 and Extended Data Fig. 5).

4. Fig. 3e-Can the N355S mutation confer resistance to Rta or is resistance mediated solely by L362P? The N355S mutation was not tested on its own. It is curious that there are two mutations in the tape measure protein if they both confer the same resistance to Rta. This is total speculation, but if N355S does not confer resistance to the Rta of PLE11, perhaps it confers resistance to a different Rta that these phages have encountered.

As the single N355S substitution was not present on its own in clinical isolates, we did not initially test it for sufficiency in mediating escape from Rta. But in response to this comment, and to be thorough, we have now reverted the TMP substitution(s) in clinical phages and in the process generated a phage with only the N355S substitution. Phage with the N355S (but not L362P) are still restricted by Rta and PLE11 (these new results are now presented in Extended Data fig 6b and c), indicating that the L362P substitution is the necessary and sufficient substitution in the clinical isolates to evade Rta/PLE11.

We have made the following edit (in italics)

To confirm the functional role of naturally evolved TMP substitutions, we engineered L362P or L362P + N355S into the CRISPR-Cas(+) Rta-sensitive historical isolate of ICP1 used in experimental evolution. Both variants were sufficient to evade phage defense mediated by PLE11 via Rta (Fig. 3e and f). *Further, reversion of these substitutions in clinical ICP1 demonstrated that L362P was the necessary and sufficient substitution to evade PLE11 and Rta-mediated restriction (Extended Data Fig. 6b and c), demonstrating that ICP1 evolved in nature under selection pressures imposed by a novel phage defense in V. cholerae.*

5. If a PLE11+ host was infected by a CRISPR-Cas ICP-1 phage, then the Rta from PLE11 would restrict ICP-1 but the CRISPR-Cas would restrict propagation of the PLE11. The authors

hypothesize that this mechanism allows vertical transmission of the PLE11 and thus can be selected for. This is an interesting idea, and I think it could be better noted in Fig. 4h to emphasize that in this case the PLE11 is maintained in the genome. Along this line, would the genome copy of PLE11 be targeted by the ICP-1 CRISPR-Cas array?

We thank the reviewer for their insightful question regarding the interaction between PLE11 and CRISPR-Cas(+) ICP1 phages. This comment has helped us to clarify a key aspect of our model.

The fate of a PLE11(+) cell infected by any ICP1 phage is cell death. However, the presence of Rta provides robust protection for the overall population of *V. cholerae* by restricting phage tail assembly and thus preventing the spread of the phage to neighboring cells. This population-level protection, regardless of the phage's anti-PLE mechanism, ensures the vertical inheritance and selection of PLE11 within the clonal *V. cholerae* population. To provide experimental support for this model, we have included new data in Extended Data Fig. 9b showing that the infection of PLE11(+) *V. cholerae* with CRISPR-Cas(+) ICP1 results in DNA-filled capsids but no tailed virions dependent on Rta-mediated restriction.

Based on these points (and additional data we generated regarding the timing of Rta expression in relation to other points raised), we have revised the manuscript text to state:

Importantly, Rta is expressed prior to ICP1 infection (Extended Data Fig. 9a). Its activity therefore provides robust inhibition of ICP1's tail formation even when the PLE genome is targeted for degradation by ICP1's nucleolytic anti-PLE mechanisms (Fig. 4h and Extended Data Fig. 9b), thus providing a selective advantage to the population when PLE11(+) *V. cholerae* is infected by CRISPR-Cas(+) or Adi(+) phages.

Regarding the question of whether the genomic copy of PLE11 would be targeted by the ICP1 CRISPR array, the answer is yes. Our prior work has demonstrated that a non-excisable PLE can be overcome by CRISPR targeting while still retaining its anti-ICP1 activity (DOI: 10.1038/s41467-018-04786-5).

6. The Rta protein would be very useful for the host even in the absence of the PLE. Have the authors searched *V. cholerae* genomes (or more broadly) to see if the bacterial have acquired Rta-like proteins as a mechanism of phage defense? Given its small size it may have been overlooked.

We have conducted a thorough search for homologs of the Rta protein in bacterial genomes. Using BLASTp and tBLASTn, we identified only six homologs ranging from 80-100% identity to Rta. All of these hits were located within *Vibrio* species, specifically *V. cholerae* and one hit in *V. vulnificus*. Subsequent analysis of the gene neighborhoods revealed that all identified homologs are encoded within putative phage satellites, many of which appear to be uncharacterized PLE variants. We have included these new findings as Extended Data Fig. 3b. Consistent with our previous work identifying PLE-like satellites in non-cholera vibrios (doi: 10.1093/nar/gkac002), these findings suggest that Rta and its homologs are not a common feature of bacterial genomes but are instead confined to the phage satellite context. Our searches using PSI-BLAST did not identify any additional homologs outside of putative *Vibrio* satellites.

The text has been updated as follows:

Rta is a small 80 amino acid protein for which we could not identify *characterized homologs outside of the context of putative satellites (Extended Data Fig. 3b)*, nor ascertain a potential function using primary sequence or predicted structure.

7. The authors could use AlphaFold to try and model the interaction between Rta and the ICP-1 TMP (and PLE11 TMP). It would be interesting to see if the region they identified resistance mutants maps to a binding domain.

Tape measure proteins, in general, are known to be notoriously disordered and difficult to study structurally. The overall pDDLT is ~0.57, with the highest confidence localized to residues 300-400, resulting in a globular structural prediction. However, TMPs are expected to form extended helical structures in either hexamers or trimers that run through the phage tail; the alphafold predictions are inherently biased towards globular folds (doi:10.1038/s41589-024-01638-w) thus, we do not have confidence in the predicted structures.

Nonetheless, we attempted co-folding TMP and Rta in 1:1, 1:2, 2:1 complexes, and each model places Rta a little differently, and the placements do remain close to the higher confidence region of residues 300-400 of TMP, which generally is the region for TMP substitutions we observe experimentally. However, due to a lack of confidence in the TMP-predicted structure, we refrain from overinterpreting these predicted interactions.

8. One piece of data that is lacking is demonstrating the interaction of Rta with the ICP-1 TMP. The authors could try two-hybrid type or co-IP approaches to demonstrate interaction of Rta with the WT TMP but not the resistant mutant TMPs. Such experiments could be done with the PLE11 TMP too (presumably no interaction).

We thank the reviewer for this excellent suggestion and agree that demonstrating a direct interaction between Rta and the ICP1 TMP would strengthen our conclusions. We heavily prioritized our time during the review process to address this point through multiple experimental approaches.

We first attempted to demonstrate a direct interaction using co-immunoprecipitation with tagged Rta during ICP1 infection. Unfortunately, despite trying various tags (Strep-II/HA/V5), (+/-) crosslinker and different timepoints during infection, we were unable to detect any binding partners for Rta, including the TMP. This may suggest the interaction is either indirect or highly transient. Additionally, the inherent toxicity of expressing the TMP alone from a plasmid prevented us from pursuing two-hybrid assays.

We also attempted to use western blots to assess whether Rta's activity leads to TMP degradation over a time course of infection. However, the custom TMP antibody exhibited high cross-reactivity with other proteins in the *V. cholerae* lysate, making this experiment uninterpretable. A preliminary targeted proteomics analysis on lysates from Rta-expressing cells showed that the TMP protein was still present, but this approach requires further troubleshooting before we can confidently report these results.

Though we were unable to definitively prove a direct interaction, we have conducted new experiments to test alternative mechanisms and strengthen our model. We show that:

1. Rta functions as a protein, not a nucleic acid. We cloned a codon-shuffled *rta* gene and confirmed it provides comparable inhibition of ICP1, validating that the protein itself is the active inhibitor.

2. *Rta* does not target splicing. Our previous work showed that the ICP1 TMP is produced from spliced transcripts (16). We engineered a splicing-independent version of the ICP1 *tmp* gene. This mutant phage was also equally sensitive to *Rta*, demonstrating that *Rta*'s activity is not related to TMP splicing.

These new data have been incorporated into Extended Data Figure 3c and 3d, and we have revised the manuscript text accordingly:

Strengthening these findings, we repeated the experimental evolution approach using a phylogenetically distinct Adi(+) ICP1 isolate and obtained parallel results in which all escape mutants harbored substitutions within the TMP (Supplementary Table 4). *By codon swapping *rta* and engineering a splicing-independent *tmp* into ICP1, we further validated that *Rta*-mediated inhibition is independent of both its nucleic acid sequence and splicing of the TMP transcript (Extended Data Fig. 3c and d).*

We have also added text to the discussion to clarify our current understanding (in italics):

Notably, *Rta* is unique among PLE-encoded factors that interfere with ICP1's lifecycle in its ability to restrict phage assembly even under circumstances where the PLE genome is degraded. *We show that *Rta* is expressed prior to ICP1 infection (Extended Data Fig. 9a), enabling its activity to be independent of the fate of the PLE genome. Whether the potent activity of *Rta* is also due to prolonged protein stability or the specific nature of its targeting of ICP1's TMP remains to be elucidated, and the lack of recognizable bioinformatic signatures of *Rta* hinders predictions. Though the selection of numerous escapes localizing to a discrete region of the TMP (Fig. 3d) is suggestive of a direct interaction between ICP1's TMP and *Rta*, we were unable to detect this interaction using co-immunoprecipitation assays during phage infection.* The molecular diversity of phage defense systems is striking as evidenced by the more than one hundred defense systems discovered to date²⁴. However, the targeting of tail assembly as a defense strategy has only very recently been discovered^{25–27}, leaving our understanding of these defense mechanisms in its early stages.

We hope the reviewer can appreciate the considerable effort we put into addressing this question and that the new experimental data refine our hypothesis that *Rta* targets the TMP, even without a direct demonstration of the interaction itself.

Referee #2

This manuscript by Mathur et al. is a tour de force, expertly combining clinical surveillance with detailed phenotypic characterizations, and especially a lovely parallel of in-lab experimental evolution mimicking the environmental evolution. I loved the core conceit, am intrigued by the mechanism, and find that the claims about mechanism of *rta*, evolutionary pressure on ICP and PLE11 are all extremely well-supported, often with entirely orthologous (or at least independent) approaches.

As an expert in phage biology, I feel qualified to comment on every aspect of the manuscript.

Narratively, the flow of information was logical and the experiments well justified through the results. However, the manuscript did struggle a little – especially in the abstract and discussion – to balance the 'new anti-phage mechanism' and 'clinical surveillance/evolution' angles of the

paper. The gap in knowledge presented in the introduction is showing that (and how) phage predation drives pathogen evolution in disease contexts. For the “that” aspect of this claim, it is almost certainly true, probably largely considered true by all phage biologists, and is a little bit peripheral to the core findings of the paper (see moderate concerns) – although in this I feel I’m probably being more of a devil’s advocate than need be, given the claim is not that extraordinary, and my objections are pretty nit-picky. The ‘how’ aspect is stronger in the paper, although again falls a tiny bit short of being bullet proof. I know this sounds damning, but to me this is only a narrative concern – the findings of this paper *really* excite me as a truly wonderful pairing of surveillance and experimental data, as a new mechanism of phage resistance that raises some extremely interesting ecological questions about ‘altruism’ beyond those of traditional ABIs, etc.

I also felt the title itself sold the work short – making no mention of the experimental work or its conclusions.

Critical: No critical flaws that require a rethinking or challenging of the premise.

We thank the reviewer for the positive assessment and genuine enthusiasm for this work. Regarding the title, we are limited to 75 characters including spaces, and are currently at 69. We haven’t been successful in incorporating edits that also speak to the experimental work while keeping under the limits, but we are open to suggestions!

Major:

-There is only one experiment where I felt some experiments had been omitted – Fig 2d. I think the phage TMP mutants should be shown for contrast, as should the phages on a ple11 rather than just the pRTA? I am almost certain these experiments will show nothing (unless the transduction efficiencies – see minor comments below – are actually further apart from the wt system than is suggested by that presentation), but they are quick, easy, and potentially important.

We agree that demonstrating the production of tailed virions by the Rta-resistant phages with TMP mutations is important. As such, we have now performed these experiments and added the results to the revised manuscript. As predicted by the reviewer and our hypothesis, we show that ICP1 derivatives with escape mutations in the TMP produce tailed particles regardless of Rta expression (Extended Data Fig. 3f), supporting the conclusion that these mutations bypass Rta’s restriction of tail assembly.

Furthermore, we performed TEM analysis of virions resulting from a CRISPR-Cas(+) ICP1 infection of PLE11(+) cells (Extended Data Fig. 9b). The data show that the phage produces tailless virions, confirming that Rta’s restriction of tail assembly is effective against phages with an anti-PLE mechanism. We also included a control on a PLE11Δrta strain to further confirm that Rta is responsible for inhibiting tail formation.

We believe the new data, along with the extensive phenotypic analysis already present in the paper, address the reviewer’s concern. We already demonstrate that the phages that escape p-rta consistently phenocopy the phages that escape PLE11 (Figure 2b,c where we show escape from PLE11 or p-rta by the same phages; Figure 3b, c, e, f, and Extended Data Figures 2 and 6 where we show plaquing phenotypes on both hosts for the 75 phage isolates and the two engineered phages with the TMP substitutions in the 2006_Dha_E background). Given the shared phenotypes between PLE11 and p-rta mentioned above, and the new TEM data, we felt it was not necessary to show TEM images of virions from PLE11(+) cells infected with the escape phages.

We have updated the manuscript with the following text to reflect the new findings (in italics):

With the induction of Rta, we observed an abundance of genome-filled capsids lacking tails (Fig. 2d, Extended Data Fig. 3e). In contrast, ICP1 derivatives with escape substitutions in TMP produced tailed particles regardless of Rta expression (Extended Data Fig. 3f).

Moderate:

- I found it somewhat challenging to assess the number of 'independent' strains at a number of junctions. This goes for both phages and *V. cholerae*. For instance, while the 53 strains (Fig 1c) are presumably all from different stool samples, it's not clear to me how clonal they are, or what level of diversity they are. Whether they are 100% clonal, or all slightly different isn't fatal to any of the conclusions drawn, but I feel the current narration does obfuscate the level of independence of the replicates. I found this consistent throughout (e.g. L81, 88 ("91%"), 98 ("all co-circulating phage isolates"). Similarly "all escape mutants" (L181), or "one or more aa substitutions in the TMP" (L255) isn't really quantifying the number of independent events, especially since 'largely clonal' isn't described (L253).

We thank the reviewer for raising this important point regarding the level of independence and diversity in our phage and *V. cholerae* strains.

To address the clonality and diversity of the phages, we initially performed VipTree analyses on the sequenced phage genomes (Fig. 3c and Extended data Figure 4, which additionally incorporates genomes in our collection from prior studies). These analyses, based on the comparison of translated CDSs, show some phages are nearly indistinguishable while others (even with the same anti-PLE mechanism) are more divergent. To provide another evaluation of clonality, in the revised manuscript we calculated the intergenomic similarities for all phages sequenced in this study using VIRIDIC (Extended Data Fig. 5). This analysis accounts for genomic sequence similarities, factoring in percent identities, genome lengths, and the percentage of the genome aligned. The intergenomic similarities among Odn(+) phages range from 95.2% to 100% (16/18 phages are at 100%) over 100% of the genome aligned. The CRISPR-Cas (+) phages share 99.8% to 100% intergenomic similarity over 100% of the genomes aligned. The intergenomic similarities between Odn(+) and CRISPR-Cas(+) phages range from 93.1% to 95.6% with 90% of genome aligned.

The high level of conservation among ICP1 isolates is consistent with our previous work on a collection of 67 ICP1 isolates from different geographic locations collected over a time span of more than 20 years (doi.org/10.1146/annurev-virology-091919-072020), where we found an average of 99.3% nucleotide conservation over 92.6% of each genome's length. Between the most distantly related isolates, temporally separated by over 25 years, there is still 98.46% nucleotide identity over 78.48% of the genome, and some isolates recovered several years apart have 99.67% identity over 100% of the genome.

The 7th pandemic lineage of *V. cholerae* is broadly regarded as clonal, but our findings here, along with other studies, demonstrate there is still consequential genetic plasticity among 7th PET lineages. The SNP-based phylogenetic analysis (Figure 1) addresses the diversity of strains in our study. For a more extensive analysis, we refer the reviewer to the accompanying manuscript by Barton and Afrad et al. This study uses additional

analyses like Panaroo to analyze the pangenome of *V. cholerae* isolates during this surveillance period, which led to their discovery of PLE11.

Regarding the quantification of independent events for the escape phages (L181), Supplementary tables 3 and 4 show the escape mutations detected in each background (n=3 for the CRISPR(+) background, n=6 for the 2011 background). Similarly, for the co-circulating phages (L255) we have added a reference to Supplemental Table 2, which provides a comprehensive list of all phages from the surveillance along with their anti-PLE mechanism and any TMP substitutions. This allows readers to directly access the number of independent phage isolates.

- The nit-pick I alluded to about the narrative: I think “demonstrates” is too strong a word for L103, and in the discussion. The fact that Ple11 confers resistance is demonstrated. The fact that ICP1 evolution is driven by PLE11 is twice demonstrated; in vitro and in surveillance. The fact that Vibrio evolution is driven by ICP1 and that PLE11 ‘conferred a selective advantage to *V. cholerae* in situ’... is a claim that stretches just a little bit further. Similarly “propound fitness advantage” (L236) conferred to Vibrio isn’t... directly supported, just inferred. As a phage biologist who sees everything bacteria do as phage-driven, I am ready to believe this claim, especially given the associations between ICP1 and lessened disease states. But I think there is an alternative explanation that is possible – that all of this just reflects an arms race between PLE11 and ICP1, and reflects only selection on these. It may be that Vibrio’s fitness is drastically affected by the sensitivity to ICP1, yes, or it may be that Vibrio’s fitness is affected by PLE11 even in the absence of ICP1 (less likely, but possible), or it may be that this lineage was fitter for other reasons, happened to co-associate with PLE11, which happened to put selective pressure on ICP1, but that the reason for the better fitness in the current environment has more to do with the lineage than the PLE11 locus.

We have tempered our language as requested for L103: made the following edits: These data *support the model* that the novel MGE variant PLE11 conferred a selective advantage to *V. cholerae in situ* and provide direct evidence that phage resistance contributes to the succession of epidemic strains.

We also removed the word ‘profound’ at L236 but maintain that ‘the complete replacement of PLE(-) strains with PLE11(+) *V. cholerae* is consistent with the fitness advantage conferred by this novel mobile phage defense element.”

We appreciate the reviewer’s point that a direct link between PLE11 and a fitness advantage for *V. cholerae* is an inference, and we agree that it is possible, though unlikely, that other factors contributed to the selective sweep. However, the observed rapid succession of strains carrying an element that provides potent, experimentally validated resistance against the relevant genotypes of the predominant lytic phage is a powerful argument for its primary role in the observed population dynamics. As highlighted in the manuscript, the correlation between high ICP1 burden and reduced disease severity also strongly suggests that phage resistance would be beneficial to *V. cholerae* fitness in the clinical environment.

We have also tempered our language in the discussion accordingly:

This study provides ~~conclusive~~ *strong* evidence that predation by the lytic phage ICP1 ~~drives~~ *can contribute to* the selection of 7PET *V. cholerae*. The acquisition of PLE11 into the BD-1.2 sublineage enabled these strains to resist contemporaneous ICP1 phages, *likely* facilitating their proliferation *and is a likely*

factor contributing to the unusually devastating cholera outbreak in the spring of 2022¹⁴. Notably, PLE11 was also detected in BD-1.2 by another study (see related manuscript by Barton and Afrad et al.) ... Our results provide *genotypic and functional* evidence to support previous accounts of the assumed role of phage predation in limiting the duration and severity of cholera epidemics³⁰⁻³².

We believe that our revised language and our acknowledgement of these nuances accurately reflect the strength of our claims and the inherent limitations of studying these complex interactions in a natural setting.

Minor:

- The claims that “evidence of phage-driven selection of *V. cholerae* has not been demonstrated” is ... I understand the authors, but given the ctx-phage association, I'm not sure it's true.

We appreciate the reviewer's attention to this detail. The reviewer is correct that the lysogenic CTX phage has been demonstrated to drive the selection of epidemic *V. cholerae*. To clarify that our statement refers specifically to predatory phages, we have added the word 'lytic' to the sentence:

However, direct molecular evidence of *lytic* phage-driven selection of epidemic *V. cholerae* has not been demonstrated.

- Can the primers for Fig 1 F not detect degradation directly (e.g. amplify the CRISPR-targeted site?). This evidence is compelling that replication is interfered with, and I agree that the most likely explanation is – given the CRISPR system – nucleolytic, but it seems odd not to show cleavage.

ICP1's CRISPR-Cas system is a Type I F system so detecting cleavage directly using standard PCR is not practical given the processive degradation of the target. Although what we've shown is indirect evidence of the system's activity, DNA cleavage in vitro using purified proteins from ICP1 has been demonstrated by others (for example doi: 10.1073/pnas.2215098120).

-The LOD is fairly low (for phage work! Obviously) in your assays in general – but especially in F2 A and C. I thin I get why C is – presumably these are stocks around 10⁶ pfu/ml plated on the sensitive hosts, but expressed RTA is harder to bypass. Might be nice to show it on an empty control (pEV) as your baseline EOP of 1, rather than the evolved mutants as the EOP of 1? This would allow us to assess the relative change in sensitivity across the two panels.

The different limits of detection (LOD) in our assays are indeed a result of using phage stocks with varying titers. As shown in Figure 3e and f, where the same titers of stocks were used to evaluate the EOPs on PLE11 and p-*rta*, the expressed Rta is not inherently more difficult to bypass than PLE11.

We understand the reviewer's perspective about the way we've plotted the data, our current data presentation highlights the core finding: that PLE11 and Rta restrict wild-type phages (resulting in low EOPs), while the escape mutants overcome this restriction (with EOPs ~1). We believe this approach is the most effective way to illustrate the inhibition and the resulting evolutionary escape. With respect to the comment about

using an empty vector (permissive) control, that was indeed used to calculate these EOPs as indicated in the legends.

-Unless I am mis-remembering, Mini/Mind-Flayers (Mini is the satellite) does not hijack tails for virion assembly (L322).

We agree with the reviewer, this statement was not accurate considering MiniFlayer, though it is not clear exactly how this satellite transmits given it was only recently discovered and molecular mechanisms remain to be elucidated. We have therefore made the follow edit to the statement, and added a relevant recent reference for a review on satellites that speaks to this assertion (which discusses MiniFlayer as a novel satellite):

We have made the following edit:

All well characterized dsDNA phage satellites hijack phage tails for virion assembly¹⁸.

- I quibble a little with the claim in L322 that this is not in keeping with the paradigm – almost all the information to date is compatible with it. Capsid-modifying proteins denying the capsid to the helper phage is very similar (conceptually) to this, the difference is – as is later pointed out – that in these interferences with the helper phage, no functional satellite particles are released.

We agree that our initial phrasing was somewhat confusing and have revised the manuscript to better articulate the apparent conundrum we were trying to introduce. Our goal was to highlight a key paradox: while Rta interferes with helper phage tail assembly, the PLE still manages to ensure its own transmission by producing a functional, tailed virion. We believe the revised text now effectively sets up this point, providing the necessary context for a general audience:

Thus, our data reveal a key paradox: while PLE-encoded Rta interferes with phage tail assembly (Fig. 2), TEM of purified PLE11 virions revealed particles with modified capsids and contractile tails (Fig. 4a). This finding, combined with previous knowledge that the most extensively characterized PLE variant, PLE1, transduces in virions with contractile tails¹⁸ dependent on the ICP1 receptor⁶, indicates that PLE11 has a mechanism to assemble tailed virions in the presence of Rta.

- “Our results indicate that all PLEs encode Rta-independent mechanisms to disrupt phage tail assembly”. I am not sure where this claim is coming from – presumably supported by the data about all the tested PLEs requiring ICP tail proteins, but I don’t think the authors demonstrated that hijacking these is ‘disrupting tail assembly of ICP. At least in this paper, only Rta is demonstrated to interfere with assembly.

The reviewer makes a good point that our data do not demonstrate disruption of tail formation by the other PLEs, only modification (Figure 4c, 4g), therefore we have altered the wording:

Our results indicate that all PLEs encode Rta-independent mechanism(s) to *manipulate* phage tail assembly.

- There's a minor narrative point about altruism that I think is confounded by some of the writing – and I think may actually be a much bigger 'deal' than the authors make it out to be. In L456, the implication is that this is similar to other uses of the word altruism in phage defence (e.g. ABIs, even when encoded on a prophage, etc). As the authors correctly note, this is actually quite different, because by stopping phage spread through the population PLE11 is actively interfering with its own spread through the population, so it's not just sacrificing one copy of a PLE11 (presumably the neighbouring cell has one too – as do all the nearby ones), as is the case in most of these so-called altruistic systems, it's also interfering with the mechanism of spread of all its con-specific neighbours by preventing them from experiencing a productive helper phage infection. If prophage encoded rexAB "wins" by eradicating all T4-infected E. coli, what it leaves is plenty of E.coli lambda lysogens, and maybe even some sensitive E. coli for lambda to eat later. If rta "wins" by preventing the driving ICP1 extinct, PLE11 loses all ability to be horizontally transmitted.

We have thought quite a lot about this point; as PLE represents an extreme in terms of satellite-mediated interference of its helper, how does it not just go extinct? We believe that our current discussion, which highlights that this strategy is likely successful due to the highly clonal nature of *V. cholerae* outbreaks and the vertical inheritance of the PLE, sufficiently addresses this point within the confines of the paper. We appreciate the reviewer's thought and insight into this nuance and will keep it in mind for future work, where we may be able to dedicate more extensive discussion to this topic.

Line-by-Line:

25: "selection of ICP1 that achieves a convergent evolutionary outcome" wasn't clear this was within the patient population.

We have made the following edit (in italics):

"Using experimental evolution, we predict phage counteradaptations against PLE11 and document the eventual emergence and selection of *clinical* ICP1 that achieves a convergent evolutionary outcome."

26-28: Felt like a bit of a non-sequitur.

We thank the reviewer for their careful reading of the abstract and for pointing out the apparent disconnect. This section, which discusses tail formation, is one of the main findings of the study. We have revised the wording to clarify its relevance and connection to the other results presented in the abstract. "*Finally, we discover how PLEs balance their dependence on ICP1 tail proteins for horizontal transmission with the restriction of phage tail assembly by Rta: PLEs construct chimeric tails comprised of both MGE- and phage-encoded proteins to ensure their transmission.*"

40: What does "predominantly lytic" mean?

We believe there is a misunderstanding regarding our use of the word predominant, not predominantly. Among the several lytic phages of *V. cholerae*, ICP1 is the phage most commonly isolated from cholera patient stool samples. Therefore, we refer to as the predominant lytic phage in this endemic area.

114: This suggests an ABI mechanism, but the language is a bit jarring – and would be especially so to a non-expert. The distinction between protecting from plaquing and protecting the bacterial genome integrity matters.

To clarify, we have removed ~~protected~~ and are more explicit by stating the phenotype is protection from plaquing.

“Surprisingly, PLE11(+)-~~protected~~ *V. cholerae* blocked plaquing even when infected by phages encoding CRISPR-Cas or Adi (Fig. 1e).”

L248: “the anti-PLE mechanisms” should read “the known anti-PLE mechanisms”

Yes, corrected.

L372: “was restored significantly” provide magnitude and statistical test? Similarly, for ‘markedly below’ (L377). I actually think the ~1 log decrease from the delta TMP is smaller than I’d expect – implying either that only 90% of particles are satellite TMPs, or that in its absence, ICP1 TMP is still at least 10% ‘as good’. That’s not too shabby!

For the claim restored significantly (L372), the statistical test and significance levels are provided in the figure legend. We have added the statistical test for the comparison between the wild type and the double $\Delta rta\Delta tmp$ mutant, which is also significant.

We agree with the reviewer that it is interesting that ICP1’s TMP seems to be sufficient for some level of transduction. We believe our current phrasing accurately reflects this.

L389 should reference the figure (4g?)

Yes, this has been updated.

L463 might be a good place for a paragraph break.

We thank the reviewer for their suggestion to improve the structure of this paragraph. While we agree that a new paragraph could be created at the second point, we believe that the two points are part of a unified argument about how our data expand the current understanding of satellite biology. The use of "First" and "Second" is intended to guide the reader through two related ways our study contributes to this expanded perspective. Breaking them into separate paragraphs could disrupt this logical flow.

L493-494: This sentence felt awkward, as it seemed to imply that the divergence of this particular system is striking, relative to the hundreds of known systems. I think maybe the word ‘from’ is what threw me off, as I believe the intention was to highlight that there is an incredible diversity of systems discovered to date, but that the idea (as elaborated on the next line) of tail-interfering is relatively new and interesting.

We have made the following edit:

The molecular diversity of phage defense systems is striking, *with over 100* defense systems discovered to date.

—

As is often the case when a manuscript excites me - I have some points of idle curiosity I wanted to raise, none of which are germane to the paper. The authors should not feel any need to address these, they simply are an a testament to my engagement and interest in the work:

- If the odn-sensitivity region were added to PLE11, would this allow RTA-sensitive but Odn+ ICP to replicate?

In this case, we would expect the phage to replicate and package its genome into capsids, but still no tails would be made and added to the virions, as Rta would still act against the phage TMP. This would be an analogous result as what is seen with CR-Cas (+) phage infection results (Fig 4g, and new extended data fig. 9b). In previous work, we have added the Odn-sensitivity region to PLEs normally resistant to it and observed that it is sufficient to restore plaque formation by Odn(+) ICP1 (doi.org/10.7554/eLife.68339)

- If a CRISPR system targeted the rta gene, would this allow otherwise RTA-sensitive ICP to plaque?

In new data for the revision, we show that the Rta protein is made before infection (Extended data Fig 9a), so this suggested mechanism of overcoming Rta seems unlikely.

- I understand there's no need to test the natural N355S mutation by itself (since it never occurred on its own), I'm just curious if it has a phenotype.

Please see the comment above for the other reviewer about this phage mutant.

- The coding density around rta is suspiciously low for a phage/phage-like element. What's going on there?

This region of PLE is around the origin of replication (doi.org/10.7554/eLife.68339), which may in part explain the lack of coding density.

- Signed Review: Alexander Hynes

Referee #4 (Remarks to the Author):

A. Mathur et al convincingly provides a link between the dynamics of ICP-1 infection, the defence systems deployed against it, and the clinical consequences in the form of Cholera dynamics. The paper provides mechanistic insights into how PLE11 sidestepped ICP-1's adaptations to handle PLE1, and documents ongoing co-evolution between the phage and the satellite.

B. These mechanistic insights include the description of a novel defense pathway down to its molecular components, while also describing the implications of this evolutionary path on the epidemiological trends observed. It also suggests possible routes for future phage adaptation and makes a convincing argument that surveillance should be expanded to consider the phage and anticipate future sweeps mediated by negative frequency dependent selection (and raising the fascinating question of how low frequency diversity is maintained in the population).

C. The surveillance data are convincing in demonstrating the sweep studied in the MS, and the ongoing evolution of the elements involved. The experimental results strongly support the hypothesized mechanism driving the evolutionary and epidemic dynamics.

D. Both are appropriate

E. some of these conclusions appear to contrast with the contents of the companion paper, and illustrate the limits of our ability to robustly identify the causes of the observed dynamics. The presence of ICP1 is stated to correlate with less severe disease, while Barton et al states that the presence of PLE1 defenses also do so. Is this because of the phage or the defenses against it? Additionally, the discussion states “Bangladesh is part of a wider transmission network”, but that is in tension with the findings of Barton et al that for much of their study, Bangladesh was not. It might be worth addressing how generalizable and clinically relevant the PLE11 findings are, given the lack of transmissibility implied elsewhere. The evidence that PLE11 helped the bacteria defend against resident ICP1 is solid, even more so with the co-evolution evidence. However the claim of PLE11 is a *main* contributing factor of the outbreak itself is less secure. Could the pathogenicity resulting in the outbreak be due to some other genetic difference on which PLE11 piggybacked because it indeed conferred resistance to the phage? The fact that there were 2 separate PLE11 introductions is supportive if not definitive, but perhaps giving a more detailed breakdown of the other background genetic differences between the outbreak and non-outbreak related strains would be helpful? The results should also be discussed in the context of the data in the companion MS which suggests lower pathogenicity for PLE11 strains (fig 3a).

There is a lot to unpack in this comment, and we appreciate the reviewer’s careful assessment of our work in light of the accompanying manuscript.

Our manuscript demonstrates the selective advantage of PLE11, while the accompanying manuscript suggests PLEs come at a potential fitness cost. We agree that this point merits discussion; therefore, we have added:

Notably, PLE11 was also detected in BD-1.2 by another study (see related manuscript by Barton and Afrad et al.). *This accompanying work intriguingly suggests that carriage of PLEs may be associated with a trade-off, potentially compromising global outbreak potential. Our findings, when viewed in this context, provide critical mechanistic evidence to support the idea that the intense selective pressure from lytic phage predation may outweigh this potential cost, driving the rapid succession of V. cholerae strains in a hyperendemic environment.*

Regarding the apparent tension over whether Bangladesh is part of a wider transmission network, we do not believe this statement conflicts with the accompanying manuscript, which focuses on strains from the last 20 years, vs 7PET strains disseminating from this region since the 1960s (Mutreja et al, 2011). Analyses in the companion manuscript highlight a potential bottleneck for PLE-carrying strains; however, our previous work has detected other historical PLE variants globally (DOI: 10.1128/mbio.03088-21), indicating that strains with these elements have successfully disseminated in the past. It is too early to tell if PLE11(+) *V. cholerae* will eventually disseminate from this region. Our statement that Bangladesh is part of a wider transmission network reflects the established understanding of the global spread of cholera from the Ganges delta; therefore, we have modified the language to state: Although Bangladesh *has long been thought to be a part of a wider transmission network for cholera globally*⁵ (where the added reference is to Mutreja et al 2011).

We appreciate the reviewer's point that our assertion that PLE11 was a main contributing factor to the outbreak is an inference. We agree that it is possible, though unlikely, that other factors contributed to the selective sweep. However, the observed rapid succession of strains carrying an element that provides potent, experimentally validated resistance against the relevant genotypes of the predominant lytic phage is a powerful argument for its primary role in the observed population dynamics. As highlighted in the

manuscript, the correlation between high ICP1 burden and reduced disease severity also strongly suggests that phage resistance would be beneficial to *V. cholerae* fitness in the clinical environment. We have tempered our language accordingly:

This study provides ~~conclusive~~ **strong** evidence that predation by the lytic phage ICP1 ~~drives~~ **can contribute to** the selection of 7PET *V. cholerae*. The acquisition of PLE11 into the BD-1.2 sublineage enabled these strains to resist contemporaneous ICP1 phages, **likely** facilitating their proliferation, **and is a likely factor** contributing to the unusually devastating cholera outbreak in the spring of 2022¹⁴.

As far as a more detailed breakdown of other genetic differences of the outbreak and non-outbreak related strains, table S8 includes all the SNPs detected in the strains; PLE11 is the only major change differentiating the previously circulating BD-1.2 sublineage from the newly emerged sublineages (defined as sBD1.3.3 and sBD1.3.4 in Barton and Afrad). In the accompanying manuscript, they employ additional analyses, such as Panaroo, to analyze the pangenome of *V. cholerae* isolates during this surveillance period, which led to the discovery of PLE11 as the change differentiating newly emerged variants from previously circulating ones. However, we cannot know with 100% certainty that PLE11 was the sole driving force. We believe that our revised language accurately reflects the strength of our claims and the inherent limitations of studying microbial evolution in a natural setting.

F. In the laboratory evolution experiments, ICP1 CRISPR-Cas(+) mutants are evolved to escape PLE11 and find TMP mutations comparable to those seen in the population. Nevertheless, when comparing to the observed environmental ICP1 evolution "...our surveillance revealed the predicted shift from phages with the ineffective anti-PLE mechanism Odn to phages with CRISPR-Cas that overcomes PLE11's conserved anti-phage defenses and substitutions in the TMP to counter PLE11's unique Rta-mediated phage defense. While experimental evolution studies could not select for CRISPR-Cas(+) isolates from an Odn(+) population— due to the lack of pre-existing variation in genetically homogenous stocks—natural populations must maintain some level of genetic diversity, enabling selection of rare alleles and oscillations of the dominant genotypes." Is the idea that some ICP1 strain with both Odn(+) and CRISPR-Cas(+) evolved to lose Odn, or that a low prevalence CRISPR-Cas(+) strain gained an advantage and swept through the population? Clarification of this would be helpful. Additionally, would it be possible to perform an evolution experiment with a more diverse population to validate these claims?

Based on our surveillance data and prior work, we have only ever observed ICP1 genotypes with either Odn or CRISPR-Cas. We have not detected any phages carrying both anti-PLE mechanisms; as stated in the introduction, CRISPR-cas replaces *odn* in the same genomic locus. This observation, combined with the fact that Odn is ineffective against PLE11 while CRISPR-Cas is effective, leads us to favor the hypothesis that a pre-existing, low-prevalence CRISPR-Cas(+) ICP1 strain gained a selective advantage and swept through the population. As stated in a response to Reviewer 1, we could not discern if this was a new CRISPR-Cas+ strain emerging to dominate, or if the Odn locus was replaced by recombination; hence, we have removed our language around that claims from the manuscript. Nonetheless, for either mechanism of emergence of CRISPR in ICP1, natural populations must maintain more diversity than we gather from sampling clinical isolates, as stated in the manuscript.

We anticipate it would be possible to perform an evolution experiment to document the selection of CRISPR vs Odn; ICP1 isolates do recombine during co-infection as we've previously documented (doi: 10.7554/eLife.53200), which would necessitate careful analyses of the resulting phages unless the goal was only to document the rise of one anti-PLE mechanism over the other. Given the oscillations in nature that we document here, and previous work showing Odn(+) phage emerging to dominate over CRISPR-Cas (doi: 10.1128/mbio.03088-21), we did not feel it necessary to set up a laboratory evolution experiment.

In addition in Figure 1c, an arrow marks the first isolate harboring PLE11 within the phylogenetic tree, which is indicated to be in the Early PLE11 period. However, some isolates in the same clade appear to be PLE11(+) and found in the pre-PLE11 period. What does this mean?

Thank you for bringing our attention to this; the issue was in our color choice to differentiate pre-PLE11 from strains not in this study (for which fine scale date resolution was not present); the PLE11(+) strains in our study are all from the early or late PLE11 period. We have addressed this by making the colors more distinct between the time periods.

The association of PLE11 with a large outbreak is important, but the companion paper also asserts that PLE11 is associated with less severe disease. This is not necessarily wholly contradictory, but deserved to be addressed.

While the accompanying manuscript suggests a correlation between PLE11 and lower disease severity, we believe there are significant limitations to drawing causal links from the genomics of *V. cholerae* isolates alone. While we do not have information to analyze disease severity for the samples in our study, their samples (relevant to the surveillance period of our manuscript) were not evaluated for the presence of co-circulating phages or the susceptibility of the bacteria to those phages.

Ultimately, we agree that these two observations are not necessarily contradictory. It is plausible that PLE11 is associated with a decrease in disease severity for an infected individual, while simultaneously contributing to a more severe population-level outbreak by facilitating the rapid proliferation of a highly transmissible, phage-resistant clone. The fitness advantage conferred by phage resistance may outweigh any potential cost in pathogenicity at the population level, especially in a hyperendemic setting like the one we studied (hence the added text in relation to their manuscript as outlined above).

In light of this feedback, we were prompted to look more closely at the companion manuscript's supplemental data. While the aggregated data suggest a correlation between PLE11 and lower disease severity, when we stratify the data by the presence or absence of co-circulating ICP1, which is able to overcome PLE11 (based on the dates in our study), a more nuanced pattern emerges that we believe may align with our findings.

- During the early period of PLE11 emergence (when no CRISPR+ TMP* ICP1 were detected), 70% of PLE11(+) samples were from patients with severe disease (n=33)
- During the later period, when CRISPR+ TMP* ICP1 were circulating, this number decreased to 43% (n=23)

This was intriguing to us, but because their study does not include information on phage presence in those samples, and ours does not include data on disease severity, we feel it is inappropriate to draw definitive conclusions. Overall, our

combined results underscore the necessity of comprehensive, multilayered surveillance to fully understand the intricate factors that influence both individual disease outcomes and population-level epidemic dynamics.

G. the references are appropriate, but is relevant to a long-standing question in cholera epidemiology, as to whether disease dynamics reflect environmental factors or the evolution of the bacteria. In this case, it appears to be the evolution of the bacteria. Although the question of why phage are apparently important in some locations but not others may be a matter of ecology. It could be worthwhile citing some of this background for context.

We appreciate this comment, but face constraints with adding many more references; however, we have added a citation to a review article and updated the manuscript as follows:

In Bangladesh, the observed oscillations in the dominance of BD-1 and BD-2 lineages of *V. cholerae*¹⁵ and ICP1 genotypes with unique repertoires of anti-PLE counter-defenses¹³ are consistent with negative frequency-dependent selection dynamics and point to the maintenance of genetic diversity in under-sampled reservoirs. Such reservoirs may include asymptotically infected individuals and/or the aquatic environment, though the precise location(s) and degree of diversity within such a reservoir remain unknown. While these oscillations provide a framework for further elucidation of the evolution of 7PET *V. cholerae* and its phages, *it is challenging to predict widespread dynamics. Although Bangladesh has long been thought to be a part of a wider transmission network for cholera globally⁵, surveillance of phages in cholera stool samples is not standard, and the ecological factors²⁹ influencing the success of a given lineage may vary significantly with each transmission event. Ultimately, our findings underscore the necessity of a holistic approach that combines genomic surveillance with mechanistic insight to understand how emerging lineages successfully spread.*

H. This is an extremely well written and presented paper throughout. Congratulations are in order.

We thank the reviewer for their very kind words.

Referee #5 (Remarks to the Author):

I co-reviewed this manuscript with one of the reviewers who provided the listed reports.

Most comments below do not require responses, we thank the reviewers for their comments and support of our work.

Referees' comments:

Referee #1 (Remarks to the Author):

This manuscript did a great job addressing the reviewers' comments. Well done!

Referee #3 (Remarks to the Author):

I was quite favourable to the initial manuscript, and the authors have been thorough in their rebuttal. I am more than satisfied with the final product.

On my single major concern at the time was asking for them to show the TMP mutants EMs, and while I might quibble as to their placement in supplemental, I think the results themselves (expected by both myself and the authors) do a good job of crossing the proverbial Ts. The omission of the PLE versions of this is well justified by the authors.

Other reviewers raised concerns about demonstrating the direct rta-TMP interaction biochemically, but I did not feel this was lacking from the manuscript. The new efforts with a codon-shuffled, and an orthologue are nice additions, however.

The new information provided in addressing other reviewer concerns that establishes the pre-expression of RTA and its effectiveness even against anti PLE phages is a very nice addition, and also addressed one of my "idle thoughts" at the end of my review.

For the title, maybe "Validating dynamic phage-pathogen coevolution captured by clinical surveillance". Demonstrating? I don't feel strongly about it! Both are more than 6 characters. Maybe the editors will allow you leeway!

We will keep the original title which meets the requirements of the journal.

The point about the varying LOD is understandable, if a little suboptimal (it could lead the reader to think the effects are very different in magnitude in Fig 2A, C), and in my case, led to an assumption that pRTA was harder to bypass for the wt than PLE11 (which the authors note is entirely refuted by Fig 3E/F). This presentation, while suboptimal, is not misleading and is sufficient to support the claims in the manuscript.

Minor points:

Is the main body missing a header (after the summary?).

We added a 'main text' header in the final version sent for formatting to the journal, but no header will be used for this section of the text as is standard for the journal.

Referee #4 (Remarks to the Author):

Thank you for the opportunity to look at this paper and see how the authors responded. One of the other reviewers described the work as "a tour de force", which was actually a term that sprang to mind for me reviewing the previous version. The updates to the MS make it even stronger.

I'd like to thank the authors too for their thoughtful examination of the supplemental data from the accompanying MS. I agree with their comment about the value of multilayered surveillance.

Referee #5 (Remarks to the Author):

I co-reviewed this manuscript with one of the reviewers who provided the listed reports.